# Brassinosteroids control cell proliferation in the lateral root cap of the Arabidopsis root

Simon Josef Unterholzner [1,2]✉, Noemi Svolacchia [3], Tanja Mimmo[1,2], Stefano Cesco[2], Riccardo Di Mambro [4], Sabrina Sabatini [3] & Raffaele Dello Ioio [3]✉

## Abstract

Stem cell protection is key to proper plant development. The root cap tissue performs this task by enclosing delicate stem cells in the root meristem and consists of columella cells at the tip and lateral root cap (LRC) cells covering the meristem. Unlike other tissues, the formation of new layers in the LRC is balanced with a regular loss through sloughing and programmed cell death in the outermost layers. The molecular mechanisms regulating this balance are still largely unknown. Brassinosteroids (BR) hormones control meristem size, radial growth, and cell shape during root development by acting non-cell-autonomously from the LRC and the epidermis. Here, we show that BR act cell-autonomously in this tissue to control LRC cell number by promoting cell-cycle entry of LRC cells through the downstream signalling factor BRASSINAZOLE-RESISTANT-1 (BZR1). Via genetic, molecular, and pharmacological analysis, we show that BZR1 maintains the LRC cell number by promoting the expression of the cell cycle regulator *CYCLIN D3;3* (*CYCD3;3*). Our results enlighten a previously unrecognised module that ensures proper lateral root cap development.

**Keywords** Root Development; Lateral Root Cap; Brassinosteroids; BZR1; CYCD3;3
**Subject Categories** Cell Cycle; Plant Biology; Signal Transduction

## Introduction

The origin of the root represented a major innovation during vascular plant evolution, enabling anchorage and efficient absorption of water and nutrients. Root growth relies on a root meristem at the tip, wherein a population of self-renewing stem cells (called initials) continuously provides new cells for all root tissues. The root meristem comprises the stem cell niche (SCN), with distal (towards the root tip) localised root cap initials and proximal (towards the shoot) transit amplifying meristem cells (Dello Ioio et al, 2007; Dello Ioio et al, 2008; Sabatini et al, 1999). The transit

amplifying meristem cells divide a fixed number of times before exiting from the meristem and start to elongate at a boundary common to all the root tissues called the transition zone (TZ). The number of cells between the SCN and the TZ defines the meristem size (Dello Ioio et al, 2007). The root cap tissue surrounds the root meristem, with crucial protective functions for delicate meristematic cells (Arnaud et al, 2010; Berhin et al, 2019). The outermost component of the root cap, the lateral root cap (LRC), shields the meristem from environmental damage and facilitates soil penetration, partly through mucilage secretion (Arnaud et al, 2010; Kumpf and Nowack, 2015). In many species such as *Arabidopsis thaliana*, the LRC is organised in concentric layers (L1, L2, L3) (Fig. 1A) (Dolan et al, 1993): asymmetric divisions of the distal epidermis and LRC initial cell (E/LRCi) continuously produce a new LRC layer (the L1), while the oldest proximal cells in L3 undergo regulated programmed cell death (PCD), thus balancing LRC renewal with turnover and thereby maintaining proper cap size and function (Kumpf and Nowack, 2015). Several lines of evidence support the idea that the LRC can regulate meristem development in a non-cell-autonomous manner. In this tissue, cytokinin activity promotes expression of the auxin-catabolic enzyme *GRETCHEN-HAGEN3.17* (*GH3.17*) and of *PIN-FORMED 5* (*PIN5*), thereby triggering the switch from cell division to cell elongation in meristematic cells and controlling meristem development and organ growth (Di Mambro et al, 2017, 2019). Despite its importance, the molecular control of LRC homeostasis remains poorly understood. Previous studies identified a role of the four closely related NAC transcription factors FEZ, SOMBRERO (SMB), BEARSKIN1 (BRN1) and BRN2 as main regulators of development and maturation of root cap cells (Bennett et al, 2010; Kamiya et al, 2016; Willemsen et al, 2008). FEZ activity controls formative divisions in the E/LRCi cells (Willemsen et al, 2008), whereas SMB, BRN1 and BRN2 are implicated in the maturation of LRC cells in the outermost tissue layer (Bennett et al, 2010; Fendrych et al, 2014). A peptide signalling pair composed of IDA-LIKE 1 (IDL1) and its receptor HEASA-LIKE 2 (HSL2) maintains the balance between newly forming cells and detachment of the outermost layers (Shi et al, 2018). However, how layer growth in between the formation and the PCD is controlled remains vague.

The plant hormone brassinosteroids (BR) has emerged as a key regulator of root development, affecting meristem size, controlling

[1]Competence Centre for Plant Health, Free University of Bozen-Bolzano, Piazza Università, Bolzano, Italy. [2]Faculty of Agricultural, Environmental and Food Sciences, Free University of Bozen-Bolzano, Piazza Università, Bolzano, Italy. [3]Dipartimento di Biologia e Biotecnologie, Laboratory of Functional Genomics and Proteomics of Model Systems, Università di Roma, Rome, Italy. [4]Department of Biology, University of Pisa, Pisa, Italy. ✉E-mail: simonjosef.unterholzner@unibz.it; raffaele.delloioio@uniroma1.it

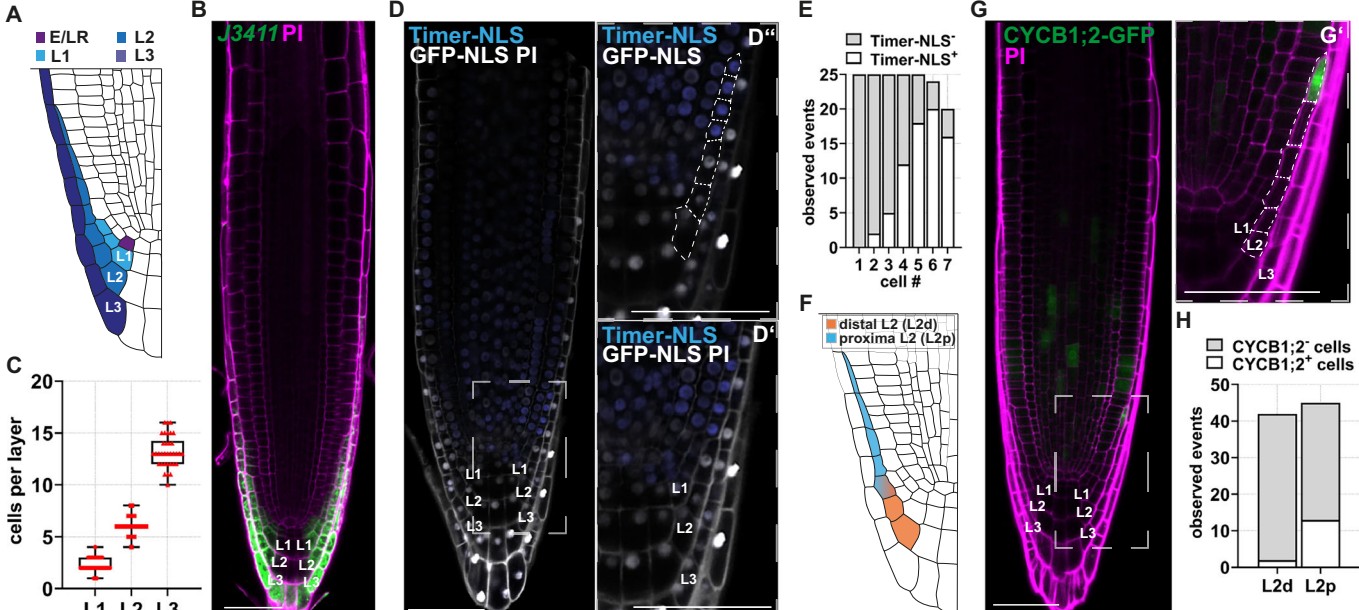

**Figure 1. Differential mitotic activity in the lateral root cap cells defines tissue homeostasis.**

(A) Diagram of the Arabidopsis wild-type roots, showing the shared epidermis/LRC stem cell (E/LRCi; purple), and the LRC layer L1 (light blue), L2 (blue) and L3 (dark blue). (B) Representative longitudinal section of the root meristem of a *J3411* plant at 6 days post-germination (dpg), showing GFP expression in the LRC cells. (C) Box plots showing the quantification of the cells per layer of Col-0 WT roots at 6 dpg. Layer 1 (L1) $n = 34$, layer 2 (L2) $n = 34$ and layer 3 (L3) $n = 34$. (D) Z-stack projection of a representative longitudinal section of the root meristem of *Timer-NLS* plants at 6 dpg. (D', D") Show a zoom of the L2 layer; in (D") only the blue and GFP channels are merged, and L2 cells highlighted with a dashed line. We observed Timer-NLS$^+$ cells in 15% of the L3 layers with a total of six Timer-NLS$^+$ out of 178 L3 cells. (E) Quantification of fade Timer-NLS positive (Timer-NLS$^+$) and Timer-NLS negative (Timer-NLS$^-$) cells along the L2 in $n = 25$ root caps. Cell #1 corresponds to the most distal L2 cells, while cell #7 the most proximal L2 cell. (F) Diagram of the Arabidopsis wild-type roots, showing the two distinct domains in the L2 layer, with the distal domain in orange (L2d) and the proximal L2 domain in blue (L2p). (G) Z-stack projection of a representative longitudinal section of the root meristem of *pCYCB1;2::CYCB1;2-GFP* plants at 6 dpg. The zoom-in section in (G') shows the L2 layer with the L2 cells highlighted. (H) Quantification of the CYCB1;2-GFP-positive (CYCB1;2$^+$) and CYCB1;2-GFP-negative (CYCB1;2$^-$) cells within the L2d and L2p domain of $n = 14$ root caps. Scale bars = 50 μm. Box plots show the median (centre line), with the box spanning the first to third quartiles (Q1–Q3); whiskers indicate the minimum (Q0) and maximum (Q4) values. Source data are available online for this figure.

meristematic division activity, tissue patterning, radial growth and cell shape (Planas-Riverola et al, 2019). The canonical BR-signalling pathway is initiated when BR are perceived by the plasma-membrane leucine-rich repeat receptor kinase BRASSINOSTEROID-INSENSITIVE 1 (BRI1) (Wang et al, 2001) and its redundant homologues BRI1-LIKE 1 (BRL1) and BRL3 (Caño-Delgado et al, 2004), which bind BR via their extracellular domain and heterodimerize with co-receptors such as BAK1/SERK3 to activate signalling (Nam and Li, 2002; Russinova et al, 2004). BR perception triggers a phosphorylation cascade leading to the inactivation of the GSK3-like kinase BRASSINOSTEROID-INSENSITIVE 2 (BIN2) and its homologue GLYCOGEN SYNTHASE KINASE 3 (GSK3) kinase family proteins, which are primary negative regulators of the BR pathway (De Rybel et al, 2009; He et al, 2002; Li et al, 2001; Rozhon et al, 2010; Yan et al, 2009). In the absence of BR, active BIN2 phosphorylates and inhibits the transcription factors of the BRASSINAZOLE-RESISTANT 1 (BZR1) and BRASSINOSTEROID-INSENSITIVE EMS SUPPRESSOR 1 (BES1) and their homologues, promoting their cytoplasmic retention and degradation (Chen et al, 2019; He et al, 2005; Planas-Riverola et al, 2019; Wang et al, 2002). BR perception and downstream activation lead to dephosphorylation of BZR1/BES1, allowing their nuclear accumulation and modulation of BR-responsive gene expression (Wang et al, 2021). BR have

been implicated in modifying the stem-cell niche by confining the expression of the QC-specifying gene *WOX5* through regulation of the MYB56 transcription factor *BRAVO* (Betegón-Putze et al, 2021; Mercadal et al, 2022; Vilarrasa-Blasi et al, 2014). Tissue-specific restoration of BR signalling of the *bri1* and *bri1 brl1 brl3* mutants revealed that BR signalling shows tissue specificity, with distinct responses in the epidermis, cortex, and vascular domains shaping local decisions on proliferation and differentiation (Chaiwanon and Wang, 2015; Fridman et al, 2014; Graeff et al, 2020; Hacham et al, 2011; Kang et al, 2017; Vragović et al, 2015). Only recently it was found that low levels of *BRI1* mRNA were also detected in multiple tissues of these rescue lines, and by using an engineered *BRI1* gene with strict tissue specific expression it was not possible to rescue root growth defects of the *bri1 brl1 brl3* mutant, suggesting that BR signalling operates in a rather cell-autonomous manner from multiple tissues of the root meristem (Blanco-Touriñán et al, 2024). Meristem-wide analyses of cell morphology have shown that BR control both the growth extent and direction by exerting tissue-specific effects on cell shape and growth anisotropy (Fridman et al, 2021; Graeff et al, 2021). In the LRC, BR signalling increases local auxin levels while dampening auxin signalling output, thereby modulating root meristem activity in a non-cell-autonomous manner (Ackerman-Lavert et al, 2021). Consistent with this model, tissue-specific knockout of *BRI1* in the epidermis and LRC leads to

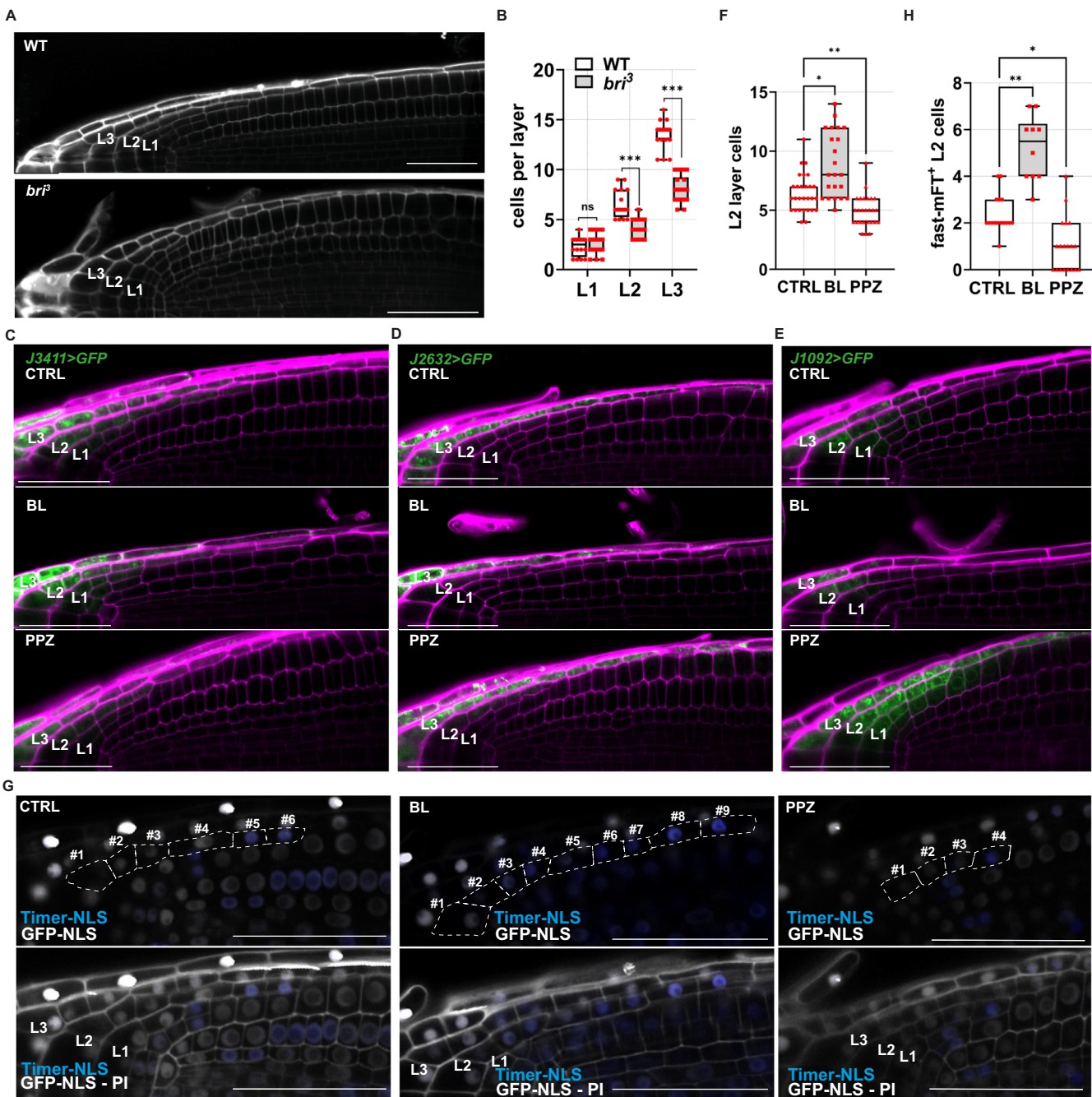

a marked reduction of the meristem size, confirming the critical contribution of BR perception in the outer tissues to overall meristem maintenance (Nolan et al, 2023). Nevertheless, because much of our understanding of tissue-specific BR function relies on *BRI1* rescue or depletion strategies, the downstream signalling components and mechanisms that confer tissue specificity remain largely unresolved (Aardening et al, 2024; Blanco-Touriñán and Hardtke, 2025).

Recently, a landmark single-cell analysis revealed that BR control the switch from proliferation to elongation cell-autonomously in the cortex, epidermis and outer tissues by promoting the expression of cell-wall-related genes via the transcription factors *HOMEOBOX*

*FROM ARABIDOPSIS THALIANA 7* (*HAT7*) and *GT-2-LIKE 1* (*GTL1*) (Nolan et al, 2023). In addition, BR signalling was shown to be dynamically coupled to the cell cycle in meristematic cells, with signalling levels dropping during mitosis and a recovery during the G1, a dynamic pattern that is required for cell cycle progression of meristem cells (Vukašinović et al, 2025). Altogether, these data support the idea that BR regulate root-meristem development both cell-autonomously and non-cell-autonomously from multiple tissues. Thus, BR signalling is not a uniform growth trigger but functions as a fine-tuned, spatially distributed network that integrates cell fate, developmental stage and chromatin context to control root growth and patterning.

**Figure 2. Brassinosteroids maintain the L2p domain to promote cell proliferation.**

(A) Representative images of root tips from wild-type (WT) and *bri³* plants at 6 dpg. LRC layers are marked as L1, L2 and L3. (B) Box plots showing the quantification of the LRC layer cells (L1, L2 and L3) in WT (*n* = 16), and *bri³* (*n* = 22) plants at 6 dpg. Pairwise comparison between WT and *bri³* LRC layers was analysed using the nonparametric two-tailed Mann–Whitney U test (exact *P* values): L1, ns*P* = 0.44; L2, ****P* < 0.001; and L3, ****P* < 0.001. (C) Representative images of root tips in *J3411* plants at 6 dpg, after 24-h of mock (CTRL), propiconazole (PPZ) or epibrassinolide (BL) treatment. PI-staining magenta, and the activation domain is marked by erGFP signal shown in green. (D) Representative images of root tips in *J2632* plants at 6 dpg, after 24-h of mock (CTRL), propiconazole (PPZ) or epibrassinolide (BL) treatment. PI-staining magenta, and the activation domain is marked by erGFP signal shown in green. (E) Representative images of root tips in *J1092* plants at 6 dpg, after 24-h of mock (CTRL), propiconazole (PPZ) or epibrassinolide ( BL) treatment. PI-staining magenta, and the activation domain is marked by erGFP signal shown in green. (F) Box plots showing the quantification of L2 layer cells in wild-type Col-0 plants treated with CTRL (*n* = 28), BL (*n* = 22), and PPZ (*n* = 26). Plants were grown for 5 days on control media and transferred for 24 h at mock (CTRL), propiconazole (PPZ) and epibrassinolide (BL). Statistical differences were calculated with the nonparametric Kruskal–Wallis test, followed by Dunn's multiple-comparisons post hoc test for pairwise comparisons between CTRL, BL and PPZ treatments (adjusted *P* values): CTRL vs. BL, **P* = 0.01; and CTRL vs. PPZ, ***P* = 0.005. (G) Z-stack projection of representative images of the root cap in *Timer-NLS* plants at 6 dpg after a 24-h mock (CTRL), propiconazole (PPZ) or epibrassinolide (BL) treatment. In the upper blue and GFP channel, the L2 cells are marked and numbered starting from the most distal cells. (H) Box plots showing the quantification of the Timer-NLS positive cells in treatments from (C), CTRL (*n* = 16), BL (*n* = 18), PPZ (*n* = 22). Statistical differences were calculated with the nonparametric Kruskal–Wallis test, followed by Dunn's multiple-comparisons post hoc test for pairwise comparisons between CTRL, +BL and PPZ treatments (adjusted *P* values): CTRL vs. BL, ***P* = 0.009; and CTRL vs. PPZ, **P* = 0.01. Scale bars = 50 μm. Box plots show the median (centre line), with the box spanning the first to third quartiles (Q1–Q3); whiskers indicate the minimum (Q0) and maximum (Q4) values. Source data are available online for this figure.

Here, we report that BR signalling acts cell-autonomously in the LRC to regulate cell number in the LRC: BZR1 promotes expression of the cell-cycle regulator *CYCLIN D3;3* (*CYCD3;3*), thereby stimulating cell-cycle entry in LRC cells and ensuring the continuous size of the mature LRC layers. This work reveals a previously unrecognised BR–BZR1–CYCD3;3 module underlying LRC size maintenance and establishes the LRC as a direct target tissue of BR signalling.

## Results and discussion

### Differential mitotic activity in lateral root cap cells defines tissue homeostasis

To gain insight into the development of the LRC, we employed the *J3411* GAL4-UAS enhancer trap line, which drives GFP expression in all LRC layers (Fig. 1B), to analyse and quantify the cells of each layer at 6 days post-germination (dpg). L1 is closest to the stem cell and is composed of about 3 cells, L2 is the middle layer and is formed of about 6 cells, L3 is in contact with the environment and is composed of a variable number of approximately 13 cells at 6 dpg (Fig. 1B,C). To investigate mechanisms supporting cell division in the LRC, we took advantage of the marker for recent cell divisions, *pKNOLLE:fast-mFluorescentTimer-NLS;pUBQ10:3xGFP-NLS* (Gaillochet et al, 2017) (hereafter referred to as *Timer-NLS*). In this line, the *KNOLLE* promoter drives the expression of a fluorescent timer protein at cytokinesis, which matures subsequently from a fade blue to a red fluorescence exhibiting form. Thus, fade blue fluorescence can be used as a proxy for recently divided cells. Analysis of the whole LRC tissue in *Timer-NLS* roots revealed that mitotic activity prevails in the L2 layer (Fig. 1D) and only occasionally cell divisions take place in the L3 (observed in 3/20 L3 layers and 6/178 L3 cells, Fig. 1D). This indicates that L2 layer cells enter the mitotic cell cycle to sustain cell proliferation-based layer growth. Interestingly, we noticed that only the proximal cells of the L2 proliferate, while the distal cells close to the stem cell niche and to the L1 remain mitotically inactive (Fig. 1D,E). Based on this observation, we subdivided the L2 into two distinct domains, a distal mitotically inactive one composed of approximately 3 cells that we named L2d and a proximal mitotically active one that we called L2p (Fig. 1F).

To corroborate this data, we analysed the cell cycle reporter *pCYCB1;2::CYCB1;2-GFP* (Boruc et al, 2010). In this line, GFP is detected only at the G2 to mitosis transition and is thereafter degraded rapidly during anaphase, thus marking only cells actively passing the G2/M phase (Boruc et al, 2010; Iwata et al, 2011). As expected, expression of this marker could be detected in the L2 layer (Fig. 1G) and again, CYCB1;2-GFP positive cells were observed mainly in the L2p (Fig. 1G,H). Altogether, these findings suggest that in the L2p domain, cells enter the mitotic cell cycle to amplify and thereby promote layer growth.

### BR control cell divisions in the L2 layer of the LRC

BR regulate meristem size and cell shape by acting non-cell-autonomously from the LRC (Ackerman-Lavert et al, 2021; Nolan et al, 2023). However, whether BR signalling also acts cell-autonomously within the LRC is unexplored. To test this possibility, we assessed whether and how BR signalling-deficient mutants affect the development of the LRC by counting the number of cells composing the LRC layers in a longitudinal section, hereafter referred to as layer size. We selected for these analyses a triple mutant line of the BR receptors, *bri1-116 brl1 brl3* (referred to as *bri³*), which is BR-insensitive (Kang et al, 2017). Individual analysis of the layer size showed that, while the L1 size is not affected in the *bri³* mutant plants, the L2 and L3 layers were significantly shorter as compared to WT plants (Fig. 2A,B). In line with these findings, the BR biosynthesis mutant *det2-1* (Fujioka et al, 1997) and the *GSK3* overexpressing line *35S::ASKθ*, which exhibits increased BZR1 phosphorylation (Rozhon et al, 2010), developed a shorter overall LRC compared with WT, with defects mainly arising in the L2 layer (Fig. EV1A–D). LRC size might be affected by altered activity of either the L2p or by altered cell death in the L3. Nonetheless, it was already reported that BR do not affect developmental programmed cell death in this tissue (Fendrych et al, 2014), suggesting that BR play a role in the L2p.

To further test this hypothesis, we treated roots with brassinolide (BL), taking advantage of the *GAL4-UAS::GFP* enhancer trap line (Haseloff, 1999) *J3411* as an LRC marker. We noticed an increased size of the L2 after treatments with BL, supporting that this hormone promotes LRC size (Fig. 2C). To understand how BR regulate LRC development, we treated with BL

the LRC-specific enhancer trap lines *J2632* and *J1092* marking, respectively, the differentiated L3 cells and the most distal cells of the L1, L2 and L3. We observed that BL did not affect the domain of *J2632* marker line, supporting that BR do not affect the L3 domain (Fig. 2D). In contrast, BL treatment restricted GFP expression in *J1092* plants, suggesting that BR limit the size of the L2d domain (Fig. 2E). To corroborate this hypothesis, we reduced BR levels by treating *J1092* plants with the BR biosynthesis inhibitor propiconazole (PPZ). PPZ-treated roots showed a broader domain of GFP expression, allowing GFP detection in the L1, L2 and L3 layers and neighbouring epidermal cells (Fig. 2E). Interestingly, we observed *J2632*-driven GFP expression also in L2 after PPZ treatment (Fig. 2D). This suggests that BR signalling controls LRC size by maintaining the mitotically active L2p domain. We reasoned that if BR promote L2p activity, we should find more cells in the L2. Thus, we quantified L2 cells after treatments with BL and PPZ. As expected, BL treatments were sufficient to trigger a ~40% increase in the number of cells in the L2 (~9 cells; Fig. 2F), while we found a significant reduction of cell number in the L2 layer of roots treated with PPZ ($P = 0.005$; Fig. 2F). Altogether, these data support the idea that BR are required for LRC development and in particular that BR are required to set the number of cells in the L2 tissue layer. As we have previously observed that the number of cells in L2 depends on cell division activity in the L2p domain, we also treated roots of plants carrying the *Timer-NLS* construct with BL and PPZ for 24 h. As expected, high BR levels were sufficient to cause a 2.2-fold increase in cell number of L2 cells (Fig. 2G,H) while inhibition of BR biosynthesis reduced cell division frequency, and, in 32% of investigated L2 cells, cell division was completely abolished (Fig. 2G,H). In line with this, PPZ treatments inhibited the G2/M marker in roots of *pCYCB1;2::CYCB1;2-GFP* (Fig. EV2A,B), while exogenous application of BL to this marker was sufficient to significantly increase the number of L2 cells expressing the G2 marker (Fig. EV2C,D). Taken together, these findings indicate that exogenous treatment of BR to the root controls LRC development by regulating the proliferative capacity of L2p cells in the L2.

## BZR1 is necessary and sufficient for cell division in the L2 layer

To gain mechanistic insights on how BR promote cell proliferation in the L2p, we questioned whether BR act cell-autonomously through the canonical BR signalling pathway. BZR1 localisation can be employed as a proxy for BR-signalling quantification, since nuclear BZR1 indicates high BR signal transduction activities while cytoplasmic localisation or absence of BZR1 suggests low BR-signalling activities (Vukašinović et al, 2025; Vukašinović et al, 2021). Therefore, to detect and quantify BR signalling in the L2, we assayed BZR1 subcellular localisation exploiting plants carrying a *35S::BZR1-CFP* construct (Rozhon et al, 2010). We observed BR signalling activity in all the L2 and L3 layers of the LRC (Fig. 3A). Nevertheless, detailed cell-by-cell quantification of nuclear BZR1-CFP in the L2 layer revealed that BZR1 concentration in the nucleus gradually increased and reached its highest levels in the L2p cells (Fig. 3A,B). Thus, within the L2 layer, BR signalling is activated in the proximal L2 corresponding to the cells with the highest cell division frequency (Fig. 1F).

We then questioned which BR-dependent transcription factor mediates the activity of this hormone in the L2p. The BZR1-type family of transcriptions factors includes six genes, the two well-studied *BZR1* and *BES1* genes and four homologues *BES1/BZR1 HOMOLOG 1* to 4 (*BEH1, BEH2, BEH3* and *BEH4*). We took advantage of a single-cell root atlas (https://rootcellatlas.org) to understand which of the BZR1-type transcription factors are expressed in the LRC tissue. Of the six genes, BZR1, BEH3 and BEH4 showed expression in LRC tissues (Fig. EV3A). In order to verify the presence of BZR1, BEH3 and BEH4 proteins in the LRC, we employed a *pBZR1::BZR1-YFP* line (Gendron et al, 2012) and generated translational reporters for *BEH3* and *BEH4* (*pBEH3::g-BEH3-eYFP* and *pBEH4::gBEH4-eYFP*). At 6-dpg, we observed BZR1-YFP in all three LRC layers (Fig. EV3B). By contrast, we did not observe expression of BEH3 in the LRC, whereas BEH4 was expressed only in proximal L3 cells (Fig. EV3C,D). Thus, we focused on BZR1 for further investigation.

Detailed analysis of the L2 showed the highest BZR1-YFP abundance in the nuclei of L2p cells (Fig. 3C,D). To understand if these cells required BZR1 to control cell number, we analysed LRC development in the loss-of-function mutant *bzr1-2* (Lachowiec et al, 2018). Interestingly, *bzr1-2* LRC development was impaired, showing a significant reduction of the cell number in the L2 layer (Fig. 3E,F), suggesting that the BZR1 protein is necessary to induce cell division in the L2, resembling BR-insensitive and deficient plants (Figs. 2A,B and EV1A,C). To assess whether BZR1 regulates mitotic activity specifically in the L2p domain, we introduced the *Timer-NLS* reporter into the *bzr1-2* mutant background. As expected, cell division in the L2p was strongly compromised in the *bzr1-2* mutant compared with WT plants, and in 46.8% of roots, division was completely abolished (Fig. 3G,H).

Next, we questioned whether BZR1 activity in the L2 is sufficient to promote cell division. To this end, we ectopically express the dominant-active and non-degradable version of *BZR1*, *bzr1-D* (Wang et al, 2002), in L2d cells using the UAS/GAL4 transactivation system. We generated *UAS::bzr1-D* plants and crossed them with the *J1092* enhancer trap line (*J1092>bzr1-D*). In these lines, GAL4 activates *bzr1-D* expression in the most distal cells of the L1, L2 (L2d) and L3 of the LRC. Analysis of *J1092>bzr1-D* roots revealed that overexpressing *bzr1-D* caused a 53% increase in the number of L2 cells compared to *J1092* control plants, suggesting that *BZR1* overexpression is sufficient to trigger cell divisions (Fig. 3I,J). To further corroborate this notion, we also generated *J1092>bin2-1* plants, where the ectopic dominant-active mutant version of *BIN2* constitutively inhibits BZR1 activity (Li et al, 2001) in L2d cells. We observed that the presence of *bin2-1* significantly reduced the number of L2 cells when compared to *J1092* (Fig. 3I,J). Furthermore, we noticed an expansion of the *J1092* domain, resembling the *J1092* domain in the BR-deficient condition. These findings were also confirmed under conditional (24 h) ectopic expression of *bzr1-D* and *bin2-1* in the L2 layer by using *pFEZ::GVG>bzr1-D* and *pFEZ::GVG>bin2-1* lines, respectively (Fig. EV3E–I). In these inducible lines, the *FEZ* promoter drives the expression in the L2 of *bzr1-D* and *bin2-1* after dexamethasone treatment (Fig. EV3E).

Similar to *bri³* (Fig. 2A,B), the LRC developmental defects in the *J1092>bin2-1* and *J1092>bzr1-D* are initiated in the L2 layer. In *J1092>bin2-1* also the cell number in the outermost L3 layer was

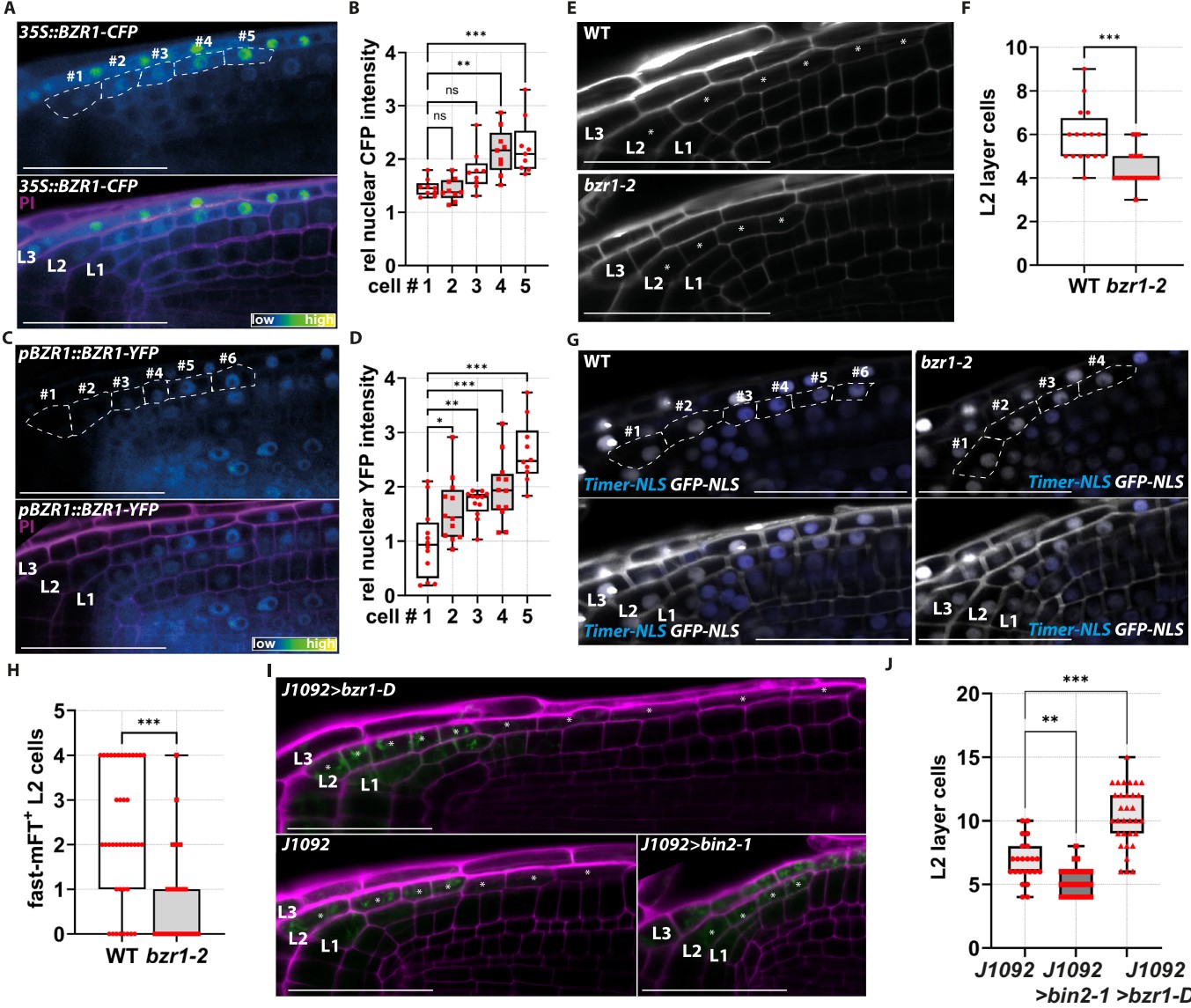

**Figure 3. Cell-autonomous BZR1 controls cell proliferation in the LRC.**

(A) Z-stack projection of a representative image of the root cap in *35S::BZR1-CFP* plants at 6 dpg. PI-staining magenta, and BZR1-CFP signal is shown as a heatmap indicated by the range indicator. In the upper CFP channel, the L2 cells are marked and numbered starting from the most distal cells. (B) Box plots showing the cell-by-cell quantification of the relative nuclear CFP (nuclear/whole cell signal intensity) from $n = 9$ L2 layers. Statistical differences between the cell #1 and cells #2-#5 were calculated by applying an ordinary one-way ANOVA followed by a post hoc Dunnett's multiple comparison test (adjusted $P$ values): cell #1 vs. cell #2, ns$P > 0.99$; cell #1 vs. cell #3 ns ($P = 0.21$); cell #1 vs. cell #4, **$P = 0.001$; and cell #1 vs. cell #5, ***$P < 0.001$. (C) Z-stack projection of a representative image of the root cap in *pBZR1::BZR1-YFP* plants at 6 dpg. PI-staining magenta, and BZR1-YFP signal is shown as a heatmap indicated by the range indicator. In the upper YFP channel, the L2 cells are marked and numbered starting from the most distal cells. (D) Box plots showing the cell-by-cell quantification of the relative nuclear YFP (nuclear/whole cell signal intensity) from $n = 12$ L2 layers. Statistical differences between cell #1 and cells #2-#5 were calculated by applying an ordinary one-way ANOVA followed by a post hoc Dunnett's multiple comparison test (adjusted P values): cell #1 vs. cell #2, ns$P > 0.99$; cell #1 vs. cell #3 ns ($P = 0.21$); cell #1 vs. cell #4, **$P = 0.001$; and cell #1 vs. cell #5, ***$P < 0.001$. (E) Representative confocal images of PI-stained root tips of wild-type (Col-0), *bzr1-2* plants at 6 dpg. Asterisks indicate the L2 cells. (F) Box plots showing the quantification of the L2 layer cells in wild-type ($n = 16$) and *bzr1-2* ($n = 16$) plants at 6 dpg. Pairwise comparison between WT and *bzr1-2* L2 layer cells was analysed using the nonparametric two-tailed Mann–Whitney $U$ test (exact $P$ value): WT vs. *bzr1-2*, ***$P < 0.001$. (G) Z-stack projection of representative images of the root cap of *Timer-NLS* wild-type (WT) and *bzr1-2 Timer-NLS* plants at 6 dpg. In the upper blue and GFP channel, the L2 cells are marked and numbered starting from the most distal cells. (H) Box plots showing the quantification of the Timer-NLS-positive cells in the experiment from (E), WT ($n = 41$), *bzr1-2* ($n = 32$). Pairwise comparison of the fast-maturing fluorescent Timer positive cells (fast-mFT+) between WT and *bzr1-2* background was analysed using the nonparametric two-tailed Mann–Whitney $U$ test (exact $P$ value): WT vs. *bzr1-2*, ***$P < 0.001$. (I) Representative confocal images of PI-stained root tips of *J1092*, *J1092>bzr1-D* and *J1092>bin2-1* plants at 6 dpg. Asterisks indicate L2 cells. PI, purple; erGFP, green. (J) Box plots showing the quantification of the L2 layer cells in *J1092* ($n = 23$), *J1092>bzr1-D* ($n = 32$) and *J1092>bin2-1* ($n = 36$). Statistical differences were calculated with the nonparametric Kruskal–Wallis test, followed by Dunn's multiple-comparisons post hoc test for pairwise comparisons (adjusted $P$ values): *J1092* vs. *J1092>bin2-1*, **$P = 0.007$; and *J1092* vs. *J1092>bzr1-D*, ***$P < 0.001$. Scale bars = 50 μm. Box plots show the median (centre line), with the box spanning the first to third quartiles (Q1–Q3); whiskers indicate the minimum (Q0) and maximum (Q4) values. Source data are available online for this figure.

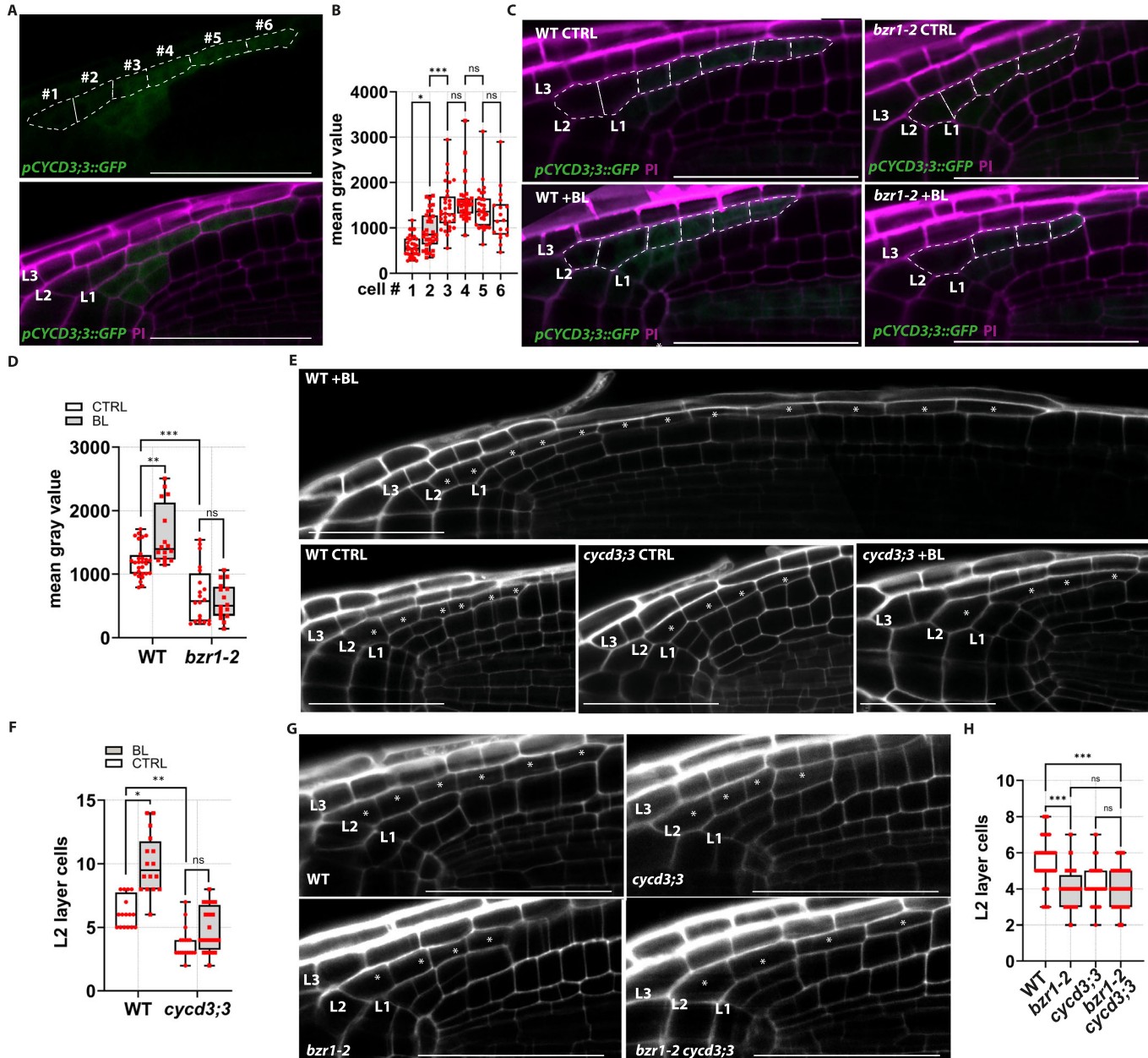

reduced compared to *J1092* (Fig. EV3J,K). Interestingly, meristem size was reduced in *J1092>bin2-1* but remained unchanged in *J1092>bzr1-D* (Fig. EV3L). Altogether, these data suggest that *BZR1* is necessary and sufficient to promote cell division in L2p cells and that BR act in these cells to control LRC development.

## BZR1 acts upstream of *CYCD3;3* to promote mitotic cell cycle entrance of L2 cells

To explore the mechanisms underlying BZR1-controlled cell divisions in the developing LRC, we searched for BZR1 direct target genes by exploiting available ChIP-Chip (data ref: Sun et al, 2010; Sun et al, 2010) and ChIP-Seq (data ref: Oh et al, 2014; Oh et al, 2014) data sets focusing on genes involved in cell cycle

regulation. Among the high-confidence BZR1 target genes, we found several genes related to the cell cycle phase transition ontology (GO:0044772). Five of them belong to the D-type cyclin gene family, a broadly conserved protein family known to form a cyclin-dependent kinase complex that phosphorylates and inactivates the RETINOBLASTOMA-RELATED PROTEIN (RBR) and thereby promotes G1-S phase transition. The cyclin-dependent kinase complex thus promotes cell cycle entrance of cells (Dewitte et al, 2007; Riou-Khamlichi et al, 1999). We noticed that *CYCD3;3* and *CYCD1;1* emerged as BZR1 targets in both BZR1 binding experiments (Fig. EV4A) and are involved in cell proliferation during embryonic and postembryonic growth (Dewitte et al, 2007; Forzani et al, 2014; Simonini et al, 2021). Among these two genes, the *CYCD3;3* gene was already reported to be expressed in the LRC

**Figure 4.  BZR1 activates *CYCD3;3* expression in the L2 cells to promote cell cycle entrance.**

(A) Representative confocal image of the root cap in p*CYCD3;3::GFP* plants at 6 dpg. PI-staining magenta, and GFP signal is shown in green. In the upper GFP channel, the L2 cells are marked and numbered starting from the most distal cells. (B) Box plots showing the cell-by-cell quantification of the mean grey value of the GFP intensity (whole cell signal intensity) in $n = 32$ L2 layers. Statistical differences between the mean grey values were assessed using an ordinary one-way ANOVA followed by Tukey's multiple-comparisons post hoc test (adjusted P values): cell #1 vs. cell #2, $*P = 0.02$; cell #2 vs. cell #3, $***P < 0.001$; cell #3 vs. cell #4, ns$P = 0.79$; cell #4 vs. cell #5, ns$P = 0.75$; and cell #5 vs. cell #6, ns$P = 0.78$. (C) Representative confocal image of the root cap from plants of p*CYCD3;3::GFP* WT and in the *bzr1-2* mutant background at 6 dpg under control conditions (CTRL) or after a 4-h treatment with 100 nM epibrassinolide ( + BL). PI-staining magenta, and GFP signal is shown in green. (D) Box plots showing the quantification of the mean grey value of the GFP intensity (whole L2 layer) from p*CYCD3;3::GFP* WT plants after CTRL ($n = 28$) and +BL ($n = 20$) treatments, and from p*CYCD3;3::GFP bzr1-2* plants after CTRL ($n = 20$) and +BL ($n = 20$) treatments. Statistical differences between mean grey values following CTRL and +BL treatment in WT and *bzr1-2* backgrounds were assessed using an ordinary one-way ANOVA followed by a Tukey's multiple-comparison post hoc test (adjusted P values): WT CTRL vs. WT + BL, $**P = 0.002$; WT CTRL vs. *bzr1-2* CTRL, $***P < 0.001$; and *bzr1-2* CTRL vs. *bzr1-2* + BL, ns$P > 0.99$. (E) Representative confocal images of PI-stained root tips of WT(Col-0), *cycd3;3* plants at 6 dpg after 24-h mock (CTRL) or epibrassinolide ( + BL) treatment. Asterisks indicate L2 cells. (F) Box plots showing the quantification of the L2 layer cells in wild-type CTRL ($n = 16$) and +BL ($n = 16$), as well as *cycd3;3* CTRL ($n = 18$) and +BL ($n = 20$). Statistical differences were calculated with the nonparametric Kruskal–Wallis test, followed by Dunn's multiple-comparisons post hoc test for pairwise comparisons (adjusted P values): WT CTRL vs. WT + BL, $*P = 0.01$; WT CTRL vs. *cycd3;3* CTRL, $**P = 0.002$; and *cycd3;3* CTRL vs. *cycd3;3* + BL, ns$P = 0.29$. (G) Representative confocal images of PI-stained root tips of WT (Col-0), *cycd3;3*, *bzr1-2* and *bzr1-2 cycd3;3* plants at 6 dpg. Asterisks indicate L2 cells. (H) Box plots showing the quantification of the L2 layer cells in WT ($n = 52$), *cycd3;3* ($n = 42$), *bzr1-2* ($n = 60$), and *bzr1-2 cycd3;3* ($n = 40$) Statistical differences were calculated with the nonparametric Kruskal–Wallis test, followed by Dunn's multiple-comparisons post hoc test for pairwise comparisons (adjusted P values): WT vs. *bzr1-2*, $***P < 0.001$; WT vs. *bzr1-2 cycd3;3*, $***P < 0.001$; *cycd3;3* vs. *bzr1-2 cycd3;3*, ns$P > 0.99$; and *bzr1-2* vs. *bzr1-2 cycd3;3*, ns$P > 0.99$. Scale bars = 50 μm. Box plots show the median (centre line), with the box spanning the first to third quartiles (Q1–Q3); whiskers indicate the minimum (Q0) and maximum (Q4) values. Source data are available online for this figure.

tissue during postembryonic root development (Forzani et al, 2014) and in a root single-cell RNA-seq experiment (Fig. EV4B). Furthermore, BR was reported to promote *CYCD3* expression in callus culture (Hu et al, 2000) and overexpression of *CYCD3;1* rescued root meristem defects in the *bri1* receptor mutant (González-García et al, 2011). We therefore hypothesised that *CYCD3;3* could be a direct BZR1 target gene in the L2 cells.

We first verified *CYCD3;3* expression in L2 cells by employing the transcriptional reporter p*CYCD3;3::GFP* (Forzani et al, 2014). We observed promoter activity in L1 and L2 cells (Fig. 4A), and a detailed cell-by-cell quantification of transcriptional signal showed the highest promoter activity in the L2p domain (Fig. 4B), consistent with the high cell division frequency and the BZR1 activity in these cells. Interestingly, exogenous BL application to p*CYCD3;3::GFP* plants resulted in an increase of *CYCD3;3* expression in the L2 layer (Figs. EV4C,D and 4C,D). To understand if BR-mediated *CYCD3;3* activation in L2 depends on BZR1, we introduced the p*CYCD3;3::GFP* reporter into the *bzr1-2* mutant background by genetic crossing. Consistent with the severely reduced mitotic activity in *bzr1-2*, we observed a reduction in p*CYCD3;3::GFP* expression in the L2 (Figs. EV4G,H and 4C,D). Furthermore, BR treatment failed to increase the *CYCD3;3* reporter signal in the *bzr1-2* background, indicating that BZR1 is required for BR to regulate *CYCD3;3* expression (Fig. 4C,D).

To further explore the control of BZR1 on *CYCD3;3* gene expression, we crossed the inducible p*FEZ:GVG>bzr1D* line with the p*CYCD3;3::GFP* line, allowing to investigate *CYCD3;3* expression upon induction of ectopic *bzr1-D* in L2d cells. Imaging of F1 progeny showed that indeed GFP signal increased following induction of ectopic *bzr1-D* expression (Fig. EV4E,F). To determine if *CYCD3;3* is required in BR-dependent L2 cell proliferation, we analysed L2 layer size in the *cycd3;3* loss-of-function mutant (Dewitte et al, 2007). *cycd3;3* roots showed a significant reduction in L2 size, and exogenous BR treatments failed to increase cell number in this layer (Fig. 4E,F). This suggested that *CYCD3;3* gene is the rate-limiting factor for BR-mediated cell proliferation. In addition, no further decrease of the L2 layer size was observed in a *bzr1-2 cycd3;3* double mutant as compared to *cycd3;3* and *bzr1-2*

single mutants (Fig. 4G,H), thus indicating that *BZR1* and *CYCD3;3* genes act in the same genetic pathway to control cell divisions during LRC development.

Our data indicate that BR, besides controlling meristem activity, radial growth and cell shape non-cell-autonomously from the LRC (Ackerman-Lavert et al, 2021; Fridman et al, 2021), also contributes directly to the development of this tissue. We propose a model where homeostasis of LRC depends not only on the interplay between formative divisions of the epidermis/LRC stem cells generating the L1 (Bennett et al, 2014; Willemsen et al, 2008) and terminal sloughing off (Shi et al, 2018) and programmed cell death (Bennett et al, 2010; Fendrych et al, 2014) in the L3, but also on the integration of the cell proliferation phase in the L2p domain. We identified two functionally distinct domains within the developing L2 layer: the distal mitotic inactive L2d domain and the proximal mitotic active L2p domain. Excitingly, BR signalling, BZR1 nuclear localisation and the onset of *CYCD3;3* expression spatially coincide in the L2p domain. Here we present genetic evidence that BR activates a BZR1-CYCD3;3 genetic module in the L2p domain that in turn is necessary for cell divisions and hence layer growth. CYCD3 D-type cyclins promote the $G_1$-to-S phase progression and maintain cells in a mitotically active state by preventing cell cycle exit (Dewitte et al, 2007). Thus, our data support that BR promote a proliferative state in L2p cells. Recently, BR signalling activity and nuclear BZR1 were shown to increase in the G1 phase, while levels drop prior to and during mitosis, coinciding with a peak in BZR1 target gene expression in the late G1 phase (Vukašinović et al, 2025). This coincides with our observation that BZR1 controls the G1-to-S phase transition by activating *CYCD3;3* expression to promote cell cycle progression. Interestingly, in dividing meristem cells, overall high BR levels or constitutive BZR1 activity promote cell elongation and differentiation (Chaiwanon and Wang, 2015; González-García et al, 2011; Hacham et al, 2011; Vukašinović et al, 2025; Vukašinović et al, 2021). In the LRC tissue, we observed high levels of BZR1 nuclear accumulation within the whole L2p domain and also ectopic expression of *bzr1-D* resulted in cell proliferation. As the *CYCD3;3* gene is known to suppress cell-cycle exit (Dewitte et al, 2007), this might explain why L2p cells remain in the mitotic

cell cycle even at constant high BZR1 levels. Future research is necessary to elucidate positional signals in the L2 that coordinate BZR1 activity to balance cell proliferation with cell differentiation and how this relates to meristem protection and function.

## Methods

### Reagents and tools table

| Reagent/resource | Reference or source | Identifier or catalogue number |
|---|---|---|
| **Experimental models** | | |
| Col-0 | | |
| *bzr1-2* | NASC | GABI_857E04 |
| *bzr1-D* | NASC | N65987 |
| *bri1-116 brl1 brl3* | Kang et al (2017) | |
| *35S::ASKθ* | Rozhon et al (2010) | |
| *det2-1* | Fujioka et al (1997) | |
| *bin2-1* | Li et al (2001) | |
| *cycd3-3* | Forzani et al (2014) | |
| *bzr1-2 cycd3;3* | This study | |
| *pBZR1::BZR1-YFP* | Gendron et al (2012) | |
| *35S::BZR1-CFP* | Rozhon et al (2010) | |
| *pBEH3::gBEH3-eYFP* | This study | |
| *pBEH4::gBEH4-eYFP* | This study | |
| *pFEZ::GVG::NLS-mCitrine* | Marquès-Bueno et al, 2016, NASC | N2106194 |
| *pCYCD3;3::GFP* | Dewitte et al (2007) | |
| *pCYCB1;2::CYCB1;2-GFP* | Boruc et al (2010) | |
| *Timer-NLS* | Gaillochet et al (2017) | |
| *bzr1-2 pCYCD3;3::GFP* | This study | |
| *bzr1-2 Timer-NLS* | This study | |
| C24 | NASC | N906 |
| *UAS::gbzr1-D* | This study | |
| *UAS::gbin2-1* | This study | |
| *J1092* | NASC | |
| *J2632* | NASC | |
| *J3411* | NASC | |
| *J1092>bin2-1* | This study | |
| *J1092>bzr1-D* | This study | |
| *pFEZ::GVG>bzr1-D* | This study | |
| *pFEZ::GVG>bin2-1* | This study | |
| **Recombinant DNA** | | |
| UAS::gbin2-1-nosT2 | This study | |
| UAS::gbzr1-D-nosT2 | This study | |

| Reagent/resource | Reference or source | Identifier or catalogue number |
|---|---|---|
| pBEH3::gBEH3-eYFP | This study | |
| pBEH4::gBEH4-eYFP | This study | |
| P4P1-UAS | Di Mambro et al (2019) | |
| pB7m43GW | Karimi et al, 2002 | |
| P2P3-eYFP | Karimi et al, 2002 | |
| P2P3-nosT2 | Karimi et al, 2002 | |
| **Oligonucleotides and other sequence-based reagents** | | |
| ATTB1_bzr1-D_fwd | GGGGACAAGTTTGTACAAAAAAGCAGGCTATGACTTCGGATGGAGCTACGTCGAC | |
| ATTB2_bzr1-D_rev | GGGGACCACTTTGTACAAGAAAGCTGGGTTCAACCACGAGCCTTCCCATTTCCAAG | |
| ATTB1_bin2-1fwd | GGGGACAAGTTTGTACAAAAAAGCAGGCTATGGCTGATGATAAGGtaaagctgct | |
| ATTB2_bin2-1_rev | GGGGACCACTTTGTACAAGAAAGCTGGGTTTAAGTTCCAGATTGATTCAAGAAGCTTAGA | |
| ATTB1_gBEH3_fwd | GGGGACAAGTTTGTACAAAAAAGCAGGCTATGACGTCGGGGACTAGAACGC | |
| ATTB2_gBEH3_rev | GGGGACCACTTTGTACAAGAAAGCTGGGTCtctggttcttgagtttccaagtgta | |
| ATTB1_gBEH4_fwd | GGGGACAAGTTTGTACAAAAAAGCAGGCTATGACATCAGGGACGAGAATGCCGAC | |
| ATTB2_gBEH4_rev | GGGGACCACTTTGTACAAGAAAGCTGGGTACCTGGTGCTTGAGTTTCCAAGAG | |
| pBEH3 fwd | agtcaatgaatataagcatagagatc | |
| pBEH3 rev | tctcagatcaccggaatttgga | |
| pBEH4 fwd | attatagcaagatttcagtgaaaaac | |
| pBEH4 rev | cactactctctgtttcttcttc | |
| bzr1-2 fwd | CATGTATTTACACGTGGACGATCG | |
| bzr1-2 rev | TGATGAAGAAGACGGGCTAACTTG | |
| LBo8474 | ATAATAACGCTGCGGACATCTACATTTT | |
| cycd3;3 fwd | TGGCTTTAGAAGAGGAGGAAGAGAG | |
| cycd3;3 rev | AACGAGATTGGAGTCACAGGATG | |
| nptII_fwd | CGCAGACCGATACCAGGATCTTG | |
| **Chemicals, enzymes and other reagents** | | |
| Epibrassinolide | Sigma-Aldrich | E1641-2MG |
| Propiconazole | Sigma-Aldrich | 45642 |
| Dexamethasone | Sigma-Aldrich | D4902 |
| Propidium Iodide | EMD Millipore | 537059 |

| Reagent/resource | Reference or source | Identifier or catalogue number |
|---|---|---|
| LR-clonase mix II | Invitrogen | 11791020 |
| BP-clonase ii enzyme mix | Invitrogen | 11789020 |
| pENTR 5'-TOPO TA Cloning Kit | Invitrogen | K59120 |
| Phosphinotricin | Duchefa | P0159 |
| **Software** | | |
| ImageJ/Fiji | https://imagej.net/software/fiji/downloads | |
| Zeiss Zen | Carl Zeiss AG | |
| Graphpad Prism 8 | https://www.graphpad.com | |
| Adobe Illustrator 2026 | Adobe | |
| **Other** | | |
| Confocal microscope Zeiss LSM800 | Carl Zeiss AG | |

## Plant materials and growth conditions

*Arabidopsis thaliana* background lines Columbia-0 (Col-0) were used as controls as the *bri³, det2-1, bzr1-2,* and *cycd3;3* are in this background. Enhancer trap lines *J3411, J1092,* and *J2632* were obtained from the NASC and are in the C24 background. Insertion sites are known for *J1092* ($+81$ bp from *AT2G8400* and $-1459$ bp from *AT2G28401*) and for J3411 ($-5801$ bp from *AT2G36360*), while it is not known for *J2632* (Radoeva et al, 2016). The *bzr1-2* mutant was obtained from the NASC collection (GABI_857E04). Homozygous mutants from the GabiKat T-DNA were identified by PCR as described (https://www.gabi-kat.de/db/primerdesign.php). *35S::ASKθ* (Rozhon et al, 2010), *35S::BZR1-CFP* (Rozhon et al, 2010), *pBZR1-BZR1-YFP* (Gendron et al, 2012), *Timer-NLS* (Gaillochet et al, 2017), *pCYCB1;2::CYCB1;2-GFP* (Boruc et al, 2010) and *pCYCD3;3::GFP* (Dewitte et al, 2007) transgenic plants have been described previously. *bzr1-2 Timer-NLS, bzr1-2 pCYCD3;3::GFP, bzr1-2 cycd3;3* were obtained by crossing. *J1092>bzr1-D, J1092>bin2-1, pFEZ::GVG>bzr1-D* and *pFEZ::GVG>bin2-1* were obtained by crossing *UAS::bzr1-D* and *UAS::bin2-1* transgenic lines with the enhancer lines. The *pFEZ:::GVG>bzr1-D; pCYCD3;3::GFP* line was obtained by crossing the *pFEZ::GVG>bzr1-D* line with the *pCYCD3;3::GFP* line and analysis were performed in F1 plants heterozygous for the two constructs.

For growth conditions, *Arabidopsis* seeds were surface sterilised, stratified at 4 °C for 2 days and seedlings were grown on half-strength Murashige and Skoog (MS) medium containing 0.8% agar and 0.3% sucrose at 22 °C in long-day conditions (16-h-light/8-h-dark cycle). L2 layer size was quantified in all experiments in 6-dpg-old roots. To have comparable LRC developmental stage (3 LRC layers), we marked germinated seeds and analysed only the roots from seeds that had germinated on the same day.

## Generation of transgenic lines

Standard molecular biology techniques and the Gateway system (Invitrogen) were used for the cloning procedures. For the

*UAS::bzr1-D* transgenic plant, the genomic sequence of *bzr1-D* (1352 bp) was amplified from genomic DNA of *Arabidopsis bzr1-D* mutant allele of *BZR1* using specific primers (ATTB1_bzr1-D_fwd/ATTB2_bzr1-D_rev) and cloned in a pDONOR221 (pDONOR221-gbzr1-D). A LR reaction was then conducted by using the pDONORP4P1-UAS, pDONOR221- gbzr1-D and a pDONORP2P3-NOSt2 vector. For the *UAS::bin2-1* transgenic plant, the genomic sequence of the *bin2-1* allele of *BIN2* (2439 bp) was amplified from genomic DNA of the Arabidopsis *bin2-1* mutant (Li et al, 2001) using specific primers (ATTB1_bin2-1fwd/ATTB2_bin2-1_rev) and cloned into a pDONOR221 (pDONOR221-gbin2-1). A LR reaction was then conducted by using the pDONORP4P1-UAS, pDONOR221-gbin2-1 and pDONORP2P3-NOSt2 vectors. The LR products were sub-cloned in the Gateway pBm43GW destination vector (Karimi et al, 2002). Plasmids were transformed into C24 background plants by floral dipping, and T2 plants were thereafter crossed into the *J1092* driver line.

For cloning the *pBEH3::gBEH3-eYFP* and *pBEH4::gBEH4-eYFP* constructs, the promoter region of *BEH3* (-2770 bp) and *BEH4* (-1213 bp) were amplified from Col-0 genomic DNA with the primer pairs pBEH3 fwd/pBEH3rev and pBEH4 fwd/pBEH4 rev and cloned into the pENTR5' using the pENTR5'-TOPO TA Cloning kit (Invitrogen). The genomic region including introns of *BEH3* and *BEH4* without the stop-codon were amplified with the primer pairs ATTB1_gBEH3_fwd/ATTB2_gBEH3_rev and ATTB1_gBEH4_fwd/ATTB2_gBEH4_rev from Col-0 genomic DNA and cloned in the pDONOR221 by BP clonase ii reaction. Final assembly of the plasmid was done with a LR reaction using the destination vector pBm43GW and a C-terminal eYFP-tag. Plasmids were transformed into Col-0 plants by floral dipping.

## Chemical treatments

In all, 5 dpg seedlings were transferred with tweezers onto solid 1/2 MS medium plates containing epibrassinolide (BL) at a final concentration of 1 nM if not otherwise indicated in the figure legend. PPZ (propiconazole) treatments were performed at a final concentration of 2 µM for 24 h, depending on the experiment (see legend). For dexamethasone (DEX) treatment 5 dpg seedlings were transferred with tweezers onto solid 1/2 MS medium plates containing DEX at a final concentration of 5 µM and were grown for an additional 24 h (see legend).

## Microscopy and image analysis

For cleared roots LRC analysis, root meristems of 6-dpg-old plants were analysed utilising a differential Interference Contrast (DIC) with Nomarski technology microscopy (Zeiss Axio Imager A2). Root tips were mounted in a chloral hydrate solution (8:3:1 mixture of chloral hydrate:water:glycerol).

Confocal images were obtained using a confocal laser scanning microscope (Zeiss LSM 800). For confocal laser scanning analysis, the cell wall was stained with 10 mg/ml propidium iodide (Sigma-Aldrich). For *35S::BZR1-CFP, pBZR1::BZR1-CFP, Timer-NLS pCYCB1;2::CYCB1;2-GFP, pCYCD3;3::GFP* and GAL4-UAS enhancer trap lines analysis, z-stack staples of the L2 layer were acquired, and the focal planes were merged into a single plane image, using the maximum projection command in ImageJ. Quantification of signal intensity was performed using the *Raw Int Den* ratio to the

area of the ROI measured with *ImageJ* as indicated in the figure legends.

## Statistical analysis

Statistical analysis was performed using GraphPad 10 (https://www.graphpad.com/scientific-software/prism/). For analyses of cell number and cell division events, nonparametric tests were applied: the Mann–Whitney *U* test for two-group comparisons and the Kruskal–Wallis test followed by Dunn's post hoc test for multiple comparisons. For analyses of fluorescence signal intensity, group comparisons were performed using one-way Welch ANOVA with Brown–Forsythe variance correction to account for unequal variances. Dunnett's post hoc test was used when comparisons were made against a single control, whereas Tukey's multiple-comparisons test was applied when all pairwise group comparisons were considered. Welch's *t* test was used for two-group comparisons. All statistical tests were two-tailed, and $p < 0.05$ was considered statistically significant.

In the box plots, the box ranges from Q1 to Q3, and the median is shown. Whiskers show the min Q0 and max Q4. The significance of the data is indicated in the graphs as ns (not significant) $P \geq 0.05$, $*P < 0.05$, $**P < 0.01$, $***P < 0.005$, (see figure legends). All experiments were performed at least $N \geq 3$ with comparable results.

## Data availability

This study includes no data deposited in external repositories.

The source data of this paper are collected in the following database record: biostudies:S-SCDT-10_1038-S44319-026-00737-0.

## Peer review information

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

## Acknowledgements

We thank Jan Lohmann, Zhiyong Wang, Jim Murray, Jenny Russinova, Christian Hardtke and Brigitte Poppenberger for sharing material. We thank Simone Egger for technical help and Luigimaria Borruso and Paula Lecca for suggestions on statistical analysis. This work was supported by a DAAD short-term fellowship (to SJU), by a mobility grant from the Autonomous Province of Bozen/Bolzano (SENSE2GROW to SJU) and by a Seal-of-excellence grant from the Autonomous Province of Bozen/Bolzano (umor-D to SJU). This study was carried out within the Agritech National Research Center and received funding from the European Union Next-GenerationEU under the Piano Nazionale di Ripresa e Resilienza (PNRR—Mission 4, Component 2, Investment 1.4; D.D. 1032 17/06/2022; CN00000022 to SS). We acknowledge financial support from the European Union Next-GenerationEU under the PNRR (Mission 4, Component 2, Investment 1.1; D.D. 1364 1/9/2023 - A conserved mechanism across species for controlling the root system architecture in rice—Grant No. P2022B4C5H, CUP C53D23007590001; to RDM), from the European Union Next-GenerationEU under the PNRR (Mission 4, Component 2, Investment 1.1, PRIN2022, Grant No. 2022NZ7M3W, CUP B53D23011370006 to SS; PRIN2022 Grant No. 20229LHY5L, CUP B53D23011210006 to RDI), and from the Giovanni Armenise FY21-22 CDA Mid-Career grant (to SS). The authors thank the Department of Innovation, Research, University and Museums of the Autonomous Province of Bozen/Bolzano for covering the Open Access publication costs.

## Author contributions

**Simon Josef Unterholzner**: Conceptualisation; Data curation; Funding acquisition; Investigation; Visualisation; Methodology; Writing—original draft; Writing—review and editing. **Noemi Svolacchia**: Investigation; Visualisation; Writing—review and editing. **Tanja Mimmo**: Resources; Funding acquisition; Writing—review and editing. **Stefano Cesco**: Funding acquisition; Writing—review and editing. **Riccardo Di Mambro**: Conceptualisation; Methodology; Writing—review and editing. **Sabrina Sabatini**: Conceptualisation; Funding acquisition; Methodology; Writing—original draft; Writing—review and editing. **Raffaele Dello Ioio**: Conceptualisation; Data curation; Investigation; Methodology; Writing—original draft; Writing—review and editing.

Source data underlying figure panels in this paper may have individual authorship assigned. Where available, figure panel/source data authorship is listed in the following database record: biostudies:S-SCDT-10_1038-S44319-026-00737-0.

## Disclosure and competing interests statement

The authors declare no competing interests.

# Expanded View Figures

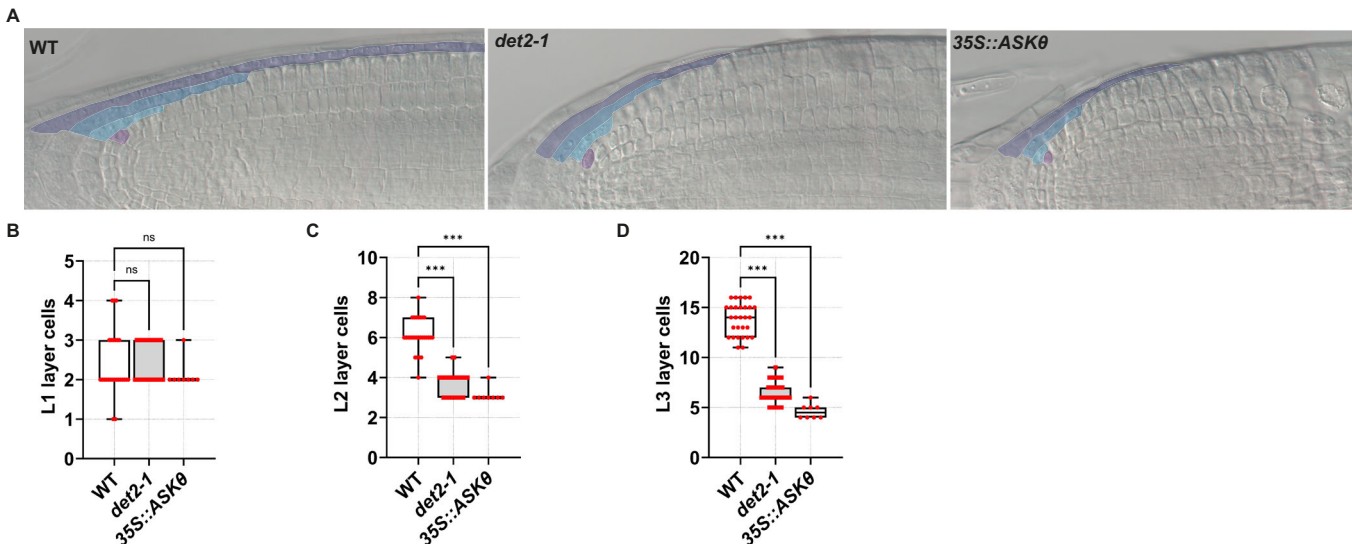

**Figure EV1. Defects in the BR biosynthesis and signalling affect LRC layer size.**

(A) Representative images of cleared root tips of wild-type (Col-0), *det2-1* and *35S::ASKθ* plants at 6 dpg. LRC layers and the stem cell are artificially coloured corresponding to the colours in (A). (B–D) Box plots showing the quantification of the L1 layer cells (B), L2 layer cells (C) and L3 layer cells (D) in wild-type (L1 $n = 26$, L2 $n = 28$, L3 $n = 29$), *det2-1* (L1 $n = 30$, L2 $n = 38$, L3 $n = 38$) and *35S::ASKθ* (L1-L3 $n = 8$) plants at 6 dpg. Statistical differences between the number of cells per layer were calculated with the nonparametric Kruskal–Wallis test, followed by Dunn's multiple-comparisons post hoc test for pairwise comparisons (adjusted P values): L1 cells in (B) WT vs. *det2-1*, ns$P = 0.78$; and WT vs. *35S::ASKθ*, ns ($P = 0.66$); L2 cells in (C) WT vs. *det2-1*, ***$P < 0.001$; and WT vs. *35S::ASKθ*, ***$P < 0.001$); L3 cells in (D) WT vs. *det2-1*, ***$P < 0.001$; and WT vs. *35S::ASKθ*, ***$P < 0.001$. Scale bars indicate 50 μm. Box plots show the median (centre line), with the box spanning the first to third quartiles (Q1–Q3); whiskers indicate the minimum (Q0) and maximum (Q4) values. Source data are available online for this figure.

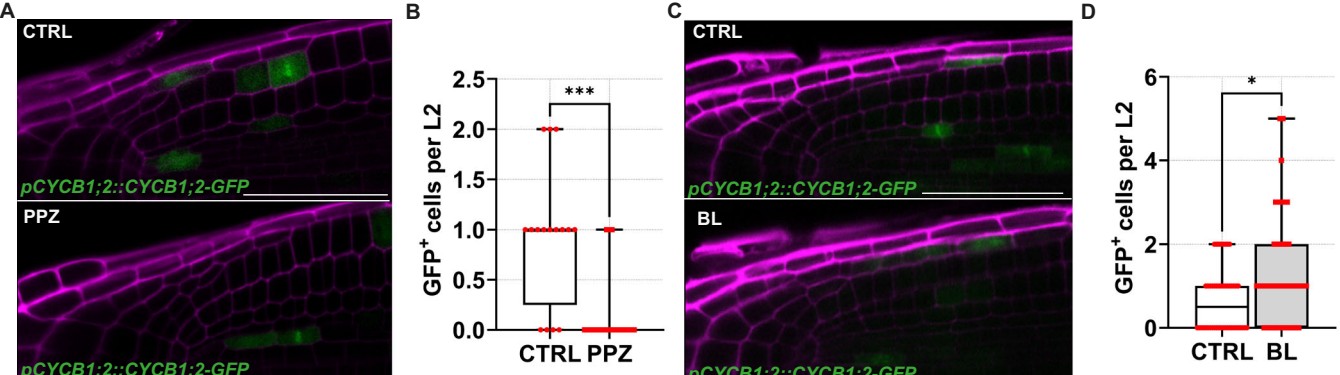

**Figure EV2. BR promote mitotic cell divisions in the L2 layer.**

(A) Z-stack projection of representative images of the root cap in *pCYCB1;2::CYCB1;2-GFP* plants at 6 dpg after a 24-h mock (CTRL) or 2 µM propiconazole (PPZ) treatment. (B) Box plots showing the quantification of CYCB1;2-GFP positive cells per L2 layer in treatments from (A). CTRL (*n* = 16) and PPZ (*n* = 22). Pairwise comparison of the CYCB1;2-GFP positive (GFP+) cells between CTRL and PPZ was analysed using the nonparametric two-tailed Mann–Whitney *U* test (exact *P* value): CTRL vs. PPZ, ***P < 0.001. (C) Z-stack projection of representative images of the root cap in *pCYCB1;2::CYCB1;2-GFP* plants at 6 dpg after 6-h mock (CTRL) or 1 nM epibrassinolide (BL) treatment. (D) Box plots showing the quantification of the CYCB1;2-GFP positive cells per L2 layer in treatments from (C). CTRL (*n* = 48) and BL (*n* = 45). Pairwise comparison of the CYCB1;2-GFP positive (GFP+) cells between CTRL and BL was analysed using the nonparametric two-tailed Mann–Whitney *U* test (exact *P* value): CTRL vs. BL, *P = 0.01. Scale bars = 50 µm. Box plots show the median (centre line), with the box spanning the first to third quartiles (Q1–Q3); whiskers indicate the minimum (Q0) and maximum (Q4) values. Source data are available online for this figure.

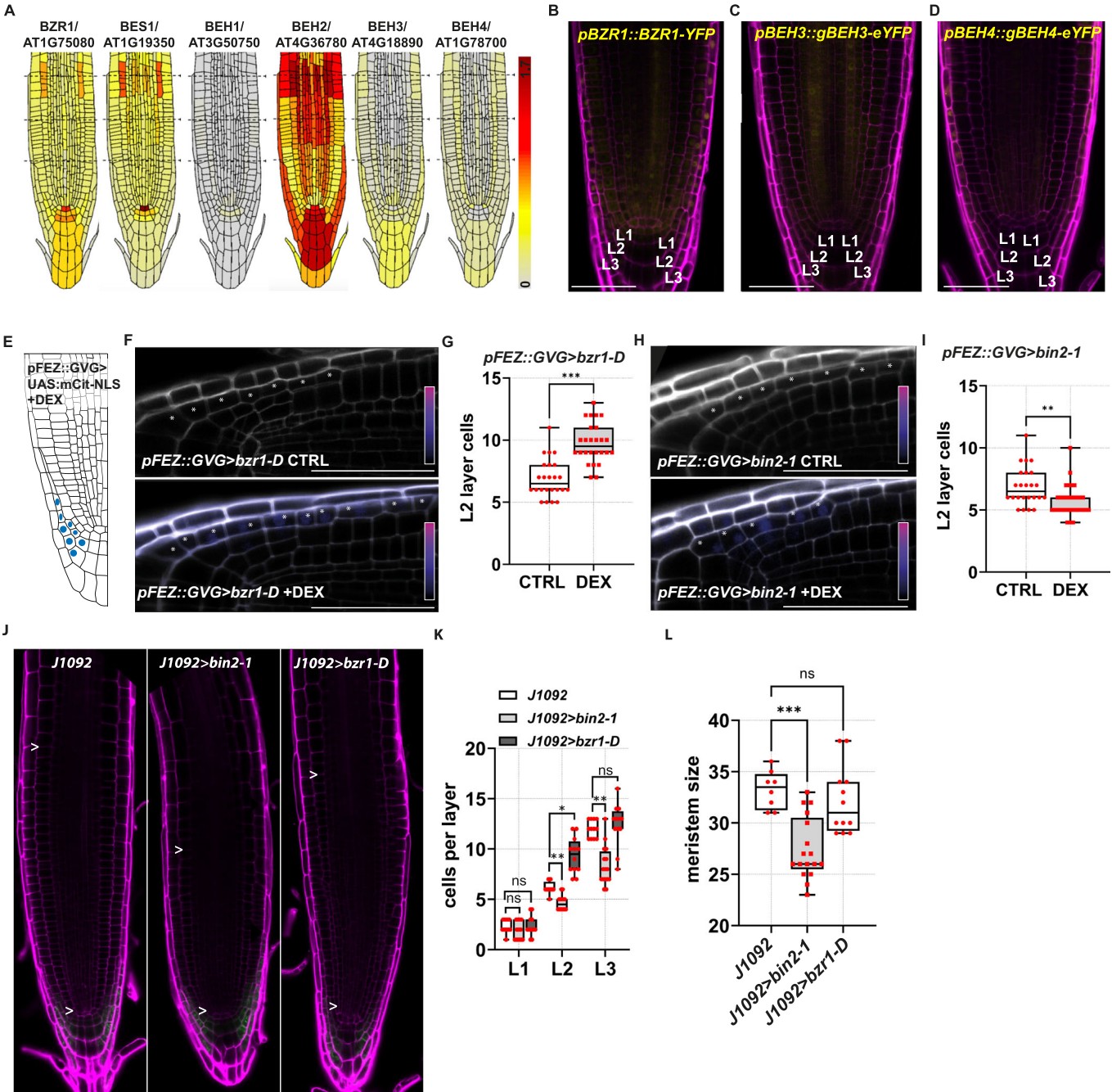

**Figure EV3. Expression patterns of BZR1-family transcription factors in the root and conditional ectopic expression of *bzr1-D* and *bin2-1* in L2 cells.**

(A) Single-cell based expression patterns of all BZR1-family transcription factors from (https://rootcellatlas.com). Expression patterns are all retrieved with the maximum threshold set at 1.7 (corresponding to the maximum value in *BES1*) for comparability, indicated by the colour scale. Note, *BEH2* expression has been reported to be induced by protoplasting (Denyer et al, 2019). (B) Z-stack projection of representative image of the root tip of *pBZR1::BZR1-YFP* plants at 6 dpg. (C) Z-stack projection of representative image of the root tip of *pBEH3::gBEH3-eYFP* plants at 6 dpg. (D) Z-stack projection of representative image of the root tip of *pBEH4::gBEH4-eYFP* plants at 6 dpg. (E) Diagram of the Arabidopsis wild-type root tip, highlighting the *pFEZ::GVG > UAS:mCit-NLS* activated cells upon DEX treatment in blue. (F) Z-stack projection of representative images of the root cap of *pFEZ::GVG>bzr1-D* mock treated plants (upper panel) and DEX treated plants (lower panel) at 6 dpg. (G) Box plots showing the quantification of the L2 layer cells *pFEZ::GVG>bzr1-D* mock treated (CTRL) plants ($n = 24$) and DEX treated (DEX) plants ($n = 26$) at 6 dpg. Pairwise comparison of the L2 layer cells between CTRL and DEX was analysed using the nonparametric two-tailed Mann–Whitney *U* test (exact *P* value): CTRL vs. DEX, ***$P < 0.001$. (H) Z-stack projection of representative images of the root cap of *pFEZ::GVG>bin2-1* mock treated plants (upper panel) and DEX treated plants (lower panel) at 6 dpg. (I) Box plots showing the quantification of the L2 layer cells *pFEZ::GVG>bin2-1* mock treated (CTRL) plants ($n = 24$) and DEX treated (DEX) plants ($n = 33$) at 6 dpg. Pairwise comparison of the L2 layer cells between CTRL and DEX was analysed using the nonparametric two-tailed Mann–Whitney *U* test (exact *P* value): CTRL vs. DEX, **$P = 0.001$. (J) Representative confocal images of PI-stained root tips of *J1092, J1092>bzr1-D and J1092>bin2-1* plants at 6 dpg. Arrowheads indicate first and the last meristematic cell in the cortex file. PI, purple; erGFP, green. (K) Box plots showing the quantification of the LRC layer cells (L1, L2 and L3) in *J1092* ($n = 12$), *J1092>bzr1-D* ($n = 12$) and *J1092>bin2-1* ($n = 16$). Statistical differences between the number of cells in each layer were calculated with the nonparametric Kruskal–Wallis test, followed by Dunn's multiple-comparisons post hoc test for pairwise comparisons between the three lines (adjusted *P* values): L1 *J1092* vs. *J1092>bin2-1*, ns ($P = 0.30$); L1 *J1092* vs. *J1092>bzr1-D*, ns$P > 0.99$; L2 *J1092* vs. *J1092>bin2-1*, **$P = 0.010$; L2 *J1092* vs. *J1092>bzr1-D*, *$P = 0.01$; L3 *J1092* vs. *J1092>bin2-1*, **$P = 0.001$; and L3 *J1092* vs. *J1092>bzr1-D*, ns$P > 0.99$. (L) Box plots showing the quantification of the meristem size (number of meristematic cortex cells) in J1092 ($n = 8$), *J1092>bzr1-D* ($n = 17$) and *J1092>bin2-1* ($n = 12$). Statistical differences between the number of meristem cells were calculated with the nonparametric Kruskal–Wallis test, followed by Dunn's multiple-comparisons post hoc test for pairwise comparisons (adjusted *P* values): J1092 vs. *J1092>bin2-1*, ***$P < 0.001$; and J1092 vs. *J1092>bzr1-D*, ns$P = 0.67$. Scale bars = 50 μm. Box plots show the median (centre line), with the box spanning the first to third quartiles (Q1–Q3); whiskers indicate the minimum (Q0) and maximum (Q4) values. Source data are available online for this figure.

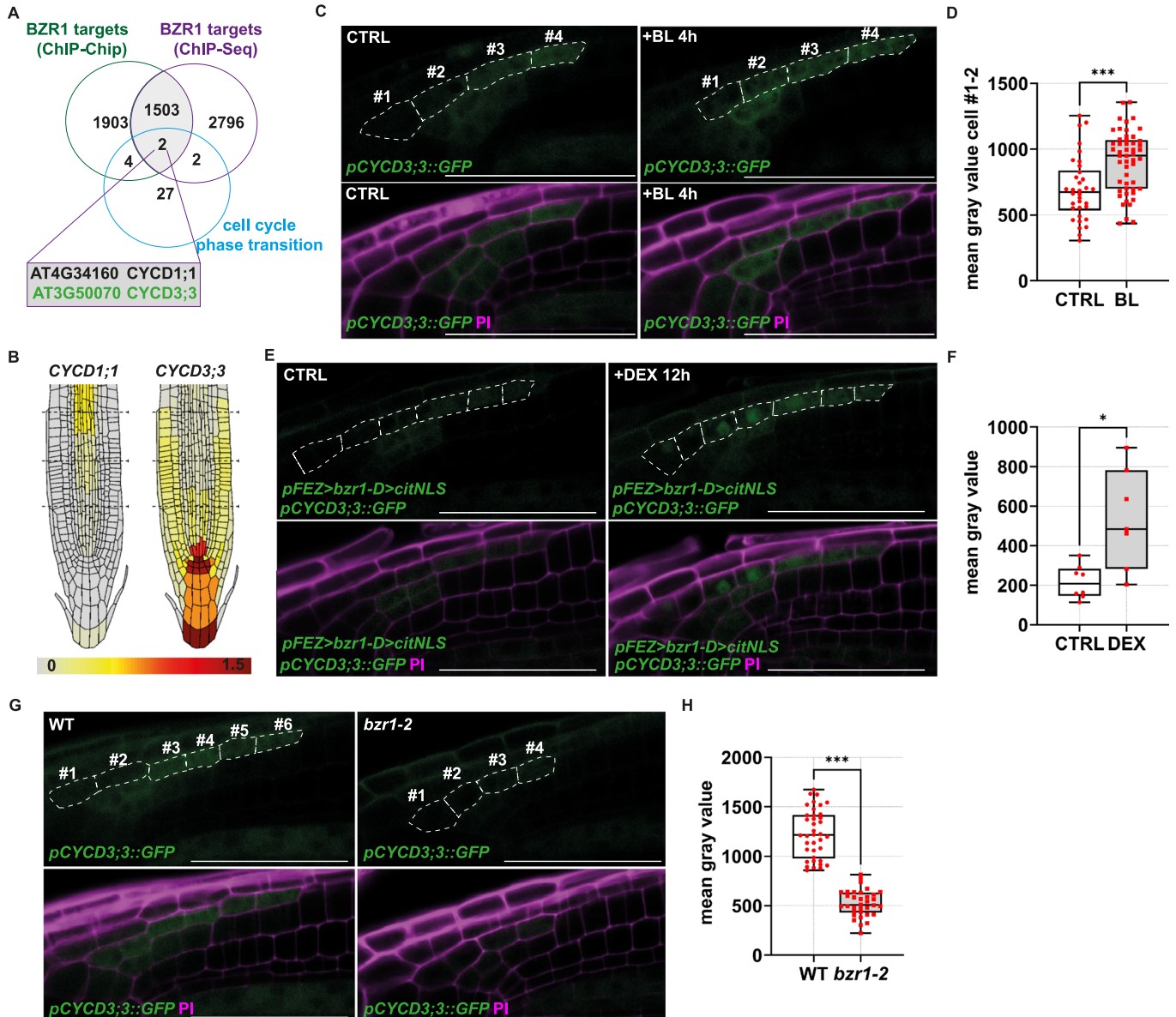

**Figure EV4. BR activates *CYCD3;3* expression in L2 cells via BZR1.**

(A) Venn diagram showing the overlap between high-confidence BZR1 target genes from a ChIP-chip experiment (data ref: Sun et al, 2010; Sun et al, 2010) and a ChIP-Seq experiment (data ref: Oh et al, 2014; Oh et al, 2014) and the gene ontology mitotic cell cycle regulation. Intersecting genes are shown in the table with the corresponding gene identifiers and naming. (B) Single-cell expression patterns of *CYCD1;1* and *CYCD3;3* genes in the root meristem tissues from (https://rootcellatlas.org). Expression patterns are all retrieved with the maximum threshold set at 1.5 (corresponding to the maximum value found for *CYCD3;3*) for comparability, as indicated in the colour scale. Note: only *CYCD3;3* mRNA was found expressed in the LRC tissue. (C) Representative confocal image of the root cap in *pCYCD3;3::GFP* wild-type plants at 6 dpg at control conditions (CTRL) or after a 4-h treatment with 100 nM epibrassinolide ( + BL). In the upper GFP channel, the L2 cells are marked and numbered starting from the most distal cells. Note that BL activates *GFP* expression mainly in cells #1 and #2 of the L2 layer. (D) Box plots showing the mean grey value quantification of GFP intensity (cell #1 and #2 of the L2 layer) in *pCYCD3;3::GFP* CTRL (n = 34) and +BL (n = 51). Statistical differences between CTRL and +BL were assessed using Welch's *t* test (*P* value): CTRL vs. +BL, ***P* < 0.001. (E) Representative confocal image of the root cap from the F1 progeny of a cross between *pFEZ>bzr1-D>citNLS* x *pCYCD3;3::GFP* at 6 dpg under control condition (CTRL) or after 12-h induction with dexamethasone ( + DEX). PI-staining magenta, and GFP signal is shown in green. (F) Box plots showing quantification of the mean grey value of GFP intensity (only cell #1 and #2 of the L2 layer) in *pFEZ>bzr1-D>citNLS* x *pCYCD3;3::GFP* CTRL (n = 8) and +DEX (n = 7). Statistical differences between CTRL and +BL were assessed using Welch's *t* test (*P* value): CTRL vs. +DEX, **P* = 0.01. (G) Representative confocal image of the root cap from *pCYCD3;3::GFP* wild-type and in the *bzr1-2* mutant background at 6 dpg. PI-staining magenta, and GFP signal is shown in green. In the upper GFP channel, the L2 cells are marked and numbered starting from the most distal cells. (H) Box plots showing the quantification of the mean grey value of GFP intensity (whole L2 layer) in *pCYCD3;3::GFP* WT (n = 38) and *pCYCD3;3::GFP bzr1-2* plants (n = 34). Statistical differences between CTRL and +BL were assessed using Welch's *t* test (*P* value): WT vs. *bzr1-2*, ***P* < 0.001. Scale bars = 50 μm. Box plots show the median (centre line), with the box spanning the first to third quartiles (Q1–Q3); whiskers indicate the minimum (Q0) and maximum (Q4) values. Source data are available online for this figure.

