## [Peer Review File · EMBO Reports]

Brassinosteroids control cell proliferation in the lateral root cap of the Arabidopsis root

Simon Unterholzner, Noemi Svolacchia, Tanja Mimmo, Stefano Cesco, Riccardo Di Mambro, Sabrina Sabatini, and Raffaele Dello Iorio

Corresponding authors: Simon Unterholzner (simonjosef.unterholzner@unibz.it) , Raffaele Dello Iorio (raffaele.delloioio@uniroma1.it)

Review Timeline:

Submission Date:	1st Aug 25
Editorial Decision:	17th Sep 25
Revision Received:	17th Dec 25
Editorial Decision:	21st Jan 26
Revision Received:	28th Jan 26
Accepted:	13th Feb 26

Transaction Report:

Dear Dr. Unterholzner

Thank you for the submission of your research manuscript to our journal. We have now received the full set of referee reports that is copied below.

As you will see, the referees acknowledge that the findings are interesting but they also note that not all conclusions are sufficiently supported by the data at hand and have a number of suggestions to further test the hypothesis brought forward. They also point out that the introduction does not accurately put your findings into context of existing literature and should be expanded. Since the detailed comments are listed below, I will not repeat them here but think that all concerns are pertinent and should be addressed.

Please let me know in case you disagree, and we can discuss the exact revision requirements further, also in a video chat, if you like.

Given these constructive comments, we would like to invite you to revise your manuscript with the understanding that the referee concerns (as detailed above and in their reports) must be fully addressed and their suggestions taken on board. Please address all referee concerns in a complete point-by-point response. Acceptance of the manuscript will depend on a positive outcome of a second round of review. It is EMBO Reports policy to allow a single round of revision only and acceptance or rejection of the manuscript will therefore depend on the completeness of your responses included in the next, final version of the manuscript.

We realize that it is difficult to revise to a specific deadline. In the interest of protecting the conceptual advance provided by the work, we recommend a revision within 3 months (December 16th). Please discuss the revision progress ahead of this time with the editor if you require more time to complete the revisions.

=====
IMPORTANT NOTE:

We perform an initial quality control of all revised manuscripts before re-review. Your manuscript will FAIL this control and the handling will be delayed IN CASE the following APPLIES:

- 1) A data availability section providing access to data deposited in public databases is missing. If you have not deposited any data, please add a sentence to the data availability section that explains that.
- 2) Your manuscript contains statistics and error bars based on $n=2$. Please use scatter blots in these cases. No statistics should be calculated if $n=2$.

=====
When submitting your revised manuscript, we will require:

- 1) a .docx formatted version of the manuscript text (including legends for main figures, EV figures and tables). Please make sure that the changes are highlighted to be clearly visible.
- 2) individual production quality figure files as .eps, .tif, .jpg (one file per figure). Please download our Figure Preparation Guidelines (figure preparation pdf) from our Author Guidelines pages <https://www.embopress.org/page/journal/14693178/authorguide> for more info on how to prepare your figures.
- 3) a .docx formatted letter INCLUDING the reviewers' reports and your detailed point-by-point responses to their comments. As part of the EMBO Press transparent editorial process, the point-by-point response is part of the Review Process File (RPF),

which will be published alongside your paper.

4) a complete author checklist, which you can download from our author guidelines (). Please insert information in the checklist that is also reflected in the manuscript. The completed author checklist will also be part of the RPF.

5) Please note that all corresponding authors are required to supply an ORCID ID for their name upon submission of a revised manuscript (). Please find instructions on how to link your ORCID ID to your account in our manuscript tracking system in our Author guidelines ()

6) We replaced Supplementary Information with Expanded View (EV) Figures and Tables that are collapsible/expandable online. A maximum of 5 EV Figures can be typeset. EV Figures should be cited as 'Figure EV1, Figure EV2' etc... in the text and their respective legends should be included in the main text after the legends of regular figures.

7) Before submitting your revision, primary datasets (and computer code, where appropriate) produced in this study need to be deposited in an appropriate public database (see < <https://www.embopress.org/page/journal/14693178/authorguide#dataavailability>>).

The accession numbers and database should be listed in a formal "Data Availability " section (placed after Materials & Method) that follows the model below (see also < <https://www.embopress.org/page/journal/14693178/authorguide#dataavailability>>). Please note that the Data Availability Section is restricted to new primary data that are part of this study.

Data availability

Additional information on source data and instruction on how to label the files are available

10) Figure legends and data quantification:

- the name of the statistical test used to generate error bars and P values,
- the EXACT p-values,
- the number (n) of independent experiments (please specify technical or biological replicates) underlying each data point,
- the nature of the bars and error bars (s.d., s.e.m.)

- If the data are obtained from n {less than or equal to} 5, show the individual data points in addition to the SD or SEM.
- If the data are obtained from n {less than or equal to} 2, use scatter blots showing the individual data points.

11) Our journal encourages inclusion of *data citations in the reference list* to directly cite datasets that were re-used and obtained from public databases. Data citations in the article text are distinct from normal bibliographical citations and should directly link to the database records from which the data can be accessed. In the main text, data citations are formatted as follows: "Data ref: Smith et al, 2001" or "Data ref: NCBI Sequence Read Archive PRJNA342805, 2017". In the Reference list, data citations must be labeled with "[DATASET]". A data reference must provide the database name, accession number/identifiers and a resolvable link to the landing page from which the data can be accessed at the end of the reference. Further instructions are available at .

12) All Materials and Methods need to be described in the main text using our 'Structured Methods' format. According to this format, the Methods section includes a Reagents and Tools Table (listing key reagents, experimental models, software and relevant equipment and including their sources and relevant identifiers) followed by a Methods and Protocols section describing the methods, ideally using a step-by-step protocol format. The aim is to facilitate adoption of the methodologies across labs. Please download and fill our Reagents and Tools Table template (.docx), which you can find in our author guidelines:

13) As part of the EMBO publication's Transparent Editorial Process, EMBO Reports publishes online a Review Process File to accompany accepted manuscripts. This File will be published in conjunction with your paper and will include the referee reports, your point-by-point response and all pertinent correspondence relating to the manuscript.

Yours sincerely,

=====

Referee #1:

Unterholzner and colleagues present a manuscript describing the role of canonical brassinosteroid signalling in driving cell divisions in the lateral root cap (LRC), which protects the root tip and stem cell niche. They provide a careful and convincing analysis of LRC growth, showing that proliferation is driven by canonical brassinosteroid signalling within a specific spatial and temporal window. By focusing on the LRC, a relatively simple system with reduced genetic complexity (brassinosteroid effects apparently largely mediated by BZR1), the authors offer clear evidence that brassinosteroid signalling can directly drive mitotic divisions - evidence that has so far been somewhat lacking due to the difficulty of separating cell division, expansion and differentiation along cell files. The data are of high quality and the experiments elegant. Nevertheless, I have a few suggestions for improvement before publication:

The framing of the study needs refinement. The presentation of the current state of the field is somewhat outdated. For example,

the view that "brassinosteroids control meristem size, radial growth and cell shape by acting cell non-autonomously from the LRC and epidermis" (lines 7, 57 & 200) has been challenged. Nolan et al. 2023 (Science) showed that brassinosteroid signalling in the cortex is also required for meristem maintenance and growth. Going even further, Blanco et al. 2024 (Science Advances) demonstrated that low receptor expression across root tissues is necessary, and that truly epidermis-restricted receptor expression is insufficient to rescue receptor mutants. These studies suggest brassinosteroid signalling acts cell-autonomously throughout the root, consistent with the authors' present finding for LRC growth.

Further regarding the literature, important papers are not mentioned. Notably work from the Delgado lab implicating a different brassinosteroid-controlled transcription factor, BRAVO, in regulating quiescent centre divisions and indirectly contributing to columella-root cap development via its interaction with WOX5 (Villarasa-Blasi et al. 2014, Developmental Cell; Betegon-Putze et al. 2021, Molecular Systems Biology); and parallel demonstrations of brassinosteroid effects on cell shape (anisotropy) (Fridman et al. 2021, Nature Plants; Graeff et al. 2021, Molecular Plant).

The use of GSK3 overexpression plants (line 63) as a proxy for 'silenced' brassinosteroid signalling is unusual and somewhat questionable, given GSK3s act in multiple pathways. Analyses of receptor mutants such as bri1 or if needed bri1 bri1 bri3 would be cleaner than the use of the GSK3 line (for which actually no verification of 'silenced' brassinosteroid signaling is presented) or even the det2 mutant and should be added.

The claim that BZR1 has "LRC-specific" expression (line 124) is misleading; it is expressed throughout the root and more strongly in the columella (but not the LRC). Expression scale differences in the root atlas also suggest redundancy with BZR1 homologues is likely. I.e. although the color gradient is similar for all the expression schemes, the scale is not: whereas the darkest color equals an expression of approximately "1" for BZR1, the same color indicates "1.6" for BES1 or "0.3" for BEH4, etc. This does not take away anything from the fact that the authors observe a matching phenotype in the bzt1 loss of function mutant (the reference #17 for this mutant is incomplete btw), but it indicates that redundancy with BZR1 homologs is still possible.

For CYCD3 fluorescence quantification, was the same line used and crossed into other backgrounds, with confirmed equal transgene copy numbers? Otherwise, results from fluorescence quantification could be misleading.

The use of Welch's t-test raises questions about variance or sample size equality.

Driver line insertion sites, if known, should be stated.

typos/grammar:

line 25: "self-renewing"?

line 70: "programmed"?

line 82: "in all cells" / "even in neighbouring"

lines 105-109: rephrase for clarity

line 154: "(SFig. 2C-F)"?

Referee #2:

This study examined the role of brassinosteroid (BR) signaling in controlling the proliferation of lateral root cap (LRC) cells in *Arabidopsis thaliana* roots. Using a series of fluorescent marker lines and genetic tools, including Timer-NLS, the authors demonstrated that a specific domain of the L2 layer of LRCs is the center of BR-mediated cell proliferation control, where CYCD3;3 plays a significant role in cell cycle progression. The study extends our understanding of the role of BRs in root meristem maintenance with higher spatial resolution. To improve the readability and scientific accuracy of the paper, I ask the authors to address the following questions.

1. Specificity of BR signaling modulation

I wonder whether the promoters used to drive bzt1-d and bin2-1 are sufficient to conclude that L2d is the site of BR signaling for LRC proliferation. Both the FEZ promoter and the J1092 driver are active outside of the L2 domain. Also, to conclude the specific role of BR signaling in the L2, the cell numbers in the L1 and L3 domains should also be quantified after the modulation of BR signaling.

2. Relationship with global root meristem (RM) regulation:

BRs are known to regulate root meristem activity and SCN maintenance. The data in Fig. 1A confirm that global modulation of BR synthesis and signaling strongly impacts RM size. Considering this, it is important to analyze whether local manipulation of BR signaling in the LRC affects RM size. If RM size remains unaffected, the significance of the postulated local LRC proliferation control becomes clearer.

3. Scientific accuracy and data quality

Please carefully check the accuracy of the introductory paragraph. The authors should clarify that the description is for

Arabidopsis thaliana roots. The explanation of the root meristem and SCN is somewhat mixed up. The number of cells in the L1-L3 layer refers only to those appearing in a longitudinal section of cell files. Each layer is composed of tens of times more cells around the circumference. The CYCD3;3 reporter fluorescent signals are too weak (Fig. 4A), and I question the reliability of the image quantification.

Miscellaneous

The text annotations in most figure panels are generally too small and have low color contrast. The graphs are also too small. Please consider labeling the L1-L3 layers in all figure panels where possible. There is mislabeling of fluorescent proteins in Figure 3A and of the X-axis in Figure 3B.

The definition of distal and proximal LRC cells is somewhat counterintuitive, considering the organ-level morphology. It would be better to mention why such a definition was adopted.

Figure 4A: If the statistical test ends with pairwise comparisons, why did the authors perform an ANOVA (Figure 4B)?

Throughout the manuscript, the term "cell division activities" would be better replaced with "cell numbers" or "cell proliferation capacities" and so on, as the authors did not measure the division activities per se.

Line 154, SFig. 3C-F should be SFig. 2C-F.

Please thoroughly check the English writing.

Referee #3:

In this exciting manuscript, the authors link BR-signalling to cell division activity of the root cap cells. The manuscript is well written, easy to read and all experiments follow in a logical order and are well designed to answer the questions at hand. I find this manuscript to contain a very interesting finding which will be of interest to a broader plant developmental biology public. There are some remarks and suggestions below to further improve the manuscript, some of which are not required, but would be nice to include in case the data would be available. The remarks regarding the statistics and inter-ecotype crosses would be the most important to address from my point of view.

Line 41: if the TIMER-NLS line changes from fade blue to red, why do we only see blue nuclei in Figure1?

Line 44: please include the penetrance of the phenotypes mentioned in text also on the figures for clarity.

Line 48: although the usage of the timer-NLS line is a nice proxy, it provides a static image of the divisions occurring in the root meristem. I am wondering why the authors did not try to catch the divisions in a time-series experiment (as is now done routinely in many labs) where one could generate a much better quantification of these events. I am not claiming this is crucial for the message the authors are bringing, but it would provide stronger quantitative evidence in case this is feasible for the authors to perform. As an alternative to the dynamic imaging, it would be good to complement the timer-NLS line by e.g. a well-known pCYCB1;1-DB-GUS/GFP line as the destruction box would also allow to highlight only the dividing cells (it is not clear to me if the line in Fig1G has a DB or not. If so, this is solved, but the authors should make this clear). My point being that it would be good to have more than a single marker to illustrate this process as it is at the basis of the manuscript.

Line 58: "However, BR mutants also display defects in the LRC tissue". This statement needs some references to support it.

Line 72: "We noticed an increased size of the L2 after treatments with BL, supporting that this hormone promotes LRC size": can this be indicated on the figure as it is perhaps not as clear to the untrained eye. Similar remark for Fig 2D and E.

Line 83: "the expression domain of J2632 was slightly expanding ". Slightly is not a very scientific term. The signal intensities in Fig 2C-E should all be quantified before making this statement based on one image. Please include the quantification in the figure to support these statements.

Line 89: Semantic discussion: the statements are made about the action of BR, but are the authors sure it is BR itself or BR-signaling which is doing this? Given this is unknown at this point in the manuscript, perhaps the statements should be about the exogenous treatment of BR to the root and not the action of BR as such.

As a more general remark relevant throughout the manuscript, the use of a t-test or ANOVA for discrete data-points (such as cell number quantifications where you only have discrete numbers: 1, 2, 3 etc; and not 1.3, 2.4, 4.5 etc) is not the best option as the data is by definition not normally distributed, but discretely. I would suggest the authors contact a statistician to apply a more correct statistical test in these cases throughout the manuscript. It would be important to ensure all results are still significant

when applying the correct statistics.

Fig 3A: panels show both YFP and CFP fusion, but the legend only mentions CFP. I am assuming the YFP in the lower panel in A should be CFP.

Line 121: 'genesand' should be 'genes and'

Line 122: "We took advantage of the single cell root atlas ", perhaps more correct to write "of a single cell atlas" as there are many published.

Line 124: looking at these predicted expression patterns, I would say none of these genes has a root cap specific expression. BZR1 is also predicted to be in the cortex. Some, if not all, of these genes are well studied. Are no markers or expression patterns published for these?

Line 141: "Hence, we generated UAS::bZR1D plants and crossed them with the J1092 enhancer trap line". Given the lines like J1092 were originally generated in the C24 ecotype, does this mean you made an inter-ecotype cross with Col-0? If so, this is known to have very striking effects on root growth and comparison to the original line in C24 is not valid. One would then need to make a control cross (J1092 x Col-0) and compare to this one. In case all was introgressed into Col-0 background, this is not required; but I did not see any info in the materials and methods regarding this. Same remark for the other crosses made involving enhancer trap lines.

Line 159: although not required, it would be nice if the authors could validate the large scale CHIP-CHIP experimental results for the BZR-CYCD3 interaction themselves. As I understand this is not a trivial experiment, it would be nice to add this data in case it is available.

Dear Editor,

We would like to thank you and all the referees for their helpful, valuable, and constructive comments on our manuscript. We have carefully revised the manuscript by addressing all issues and questions raised by the reviewers.

In particular, we have added a state-of-the-art introductory section to better place our findings within the context of the existing literature. We believe that these revisions have significantly improved the clarity and quality of the manuscript.

Please find below a detailed, point-by-point response to the reviewers' comments. All changes in the revised manuscript and our responses are highlighted in green.

With kind regards,

Simon Josef Unterholzner and Raffaele Dello Iorio

Referee #1:

Unterholzner and colleagues present a manuscript describing the role of canonical brassinosteroid signalling in driving cell divisions in the lateral root cap (LRC), which protects the root tip and stem cell niche. They provide a careful and convincing analysis of LRC growth, showing that proliferation is driven by canonical brassinosteroid signalling within a specific spatial and temporal window. By focusing on the LRC, a relatively simple system with reduced genetic complexity (brassinosteroid effects apparently largely mediated by BZR1), the authors offer clear evidence that brassinosteroid signalling can directly drive mitotic divisions - evidence that has so far been somewhat lacking due to the difficulty of separating cell division, expansion and differentiation along cell files. The data are of high quality and the experiments elegant. Nevertheless, I have a few suggestions for improvement before publication:

We thank the reviewer for their positive appreciation of our story.

The framing of the study needs refinement. The presentation of the current state of the field is somewhat outdated. For example, the view that "brassinosteroids control meristem size, radial growth and cell shape by acting cell non-autonomously from the LRC and epidermis" (lines 7, 57 & 200) has been challenged. Nolan et al. 2023 (Science) showed that brassinosteroid signalling in the cortex is also required for meristem maintenance and growth. Going even further, Blanco et al. 2024 (Science Advances) demonstrated that low receptor expression across root tissues is necessary, and that truly epidermis-restricted receptor expression is insufficient to rescue receptor mutants. These studies suggest brassinosteroid signalling acts cell-autonomously throughout the root, consistent with the authors' present finding for LRC growth.

We thank the reviewer for raising this important point and have added a complete introduction section, that includes all the most up-to-date literature on cell autonomous brassinosteroid signalling, which is also consistent with our findings.

Further regarding the literature, important papers are not mentioned. Notably work from the Delgado lab implicating a different brassinosteroid-controlled transcription factor, BRAVO, in regulating quiescent centre divisions and indirectly contributing to columella-root cap development via its interaction with WOX5 (Villarasa-Blasi et al. 2014, Developmental Cell; Betegon-Putze et al. 2021, Molecular Systems Biology); and parallel demonstrations of brassinosteroid effects on cell shape (anisotropy) (Fridman et al. 2021, Nature Plants; Graeff et al. 2021, Molecular Plant).

We thank the reviewer for the further suggested literature. We included these in the introduction section.

The use of GSK3 overexpression plants (line 63) as a proxy for 'silenced' brassinosteroid signalling is unusual and somewhat questionable, given GSK3s act in multiple pathways. Analyses of receptor mutants such as *bri1* or if needed *bri1 bri3* would be cleaner than the use of the GSK3 line (for which actually no verification of 'silenced' brassinosteroid signalling is presented) or even the *det2* mutant and should be added.

We acknowledge this valid and well-taken suggestion. Accordingly, we assessed LRC development in the *bri1-116 bri1 bri3* triple mutant (*bri³*) (Fig. 2A,B). The results are consistent with those obtained in the *det2-1* and *35S::ASK θ* lines, which have now been moved to Figure EV1, and further strengthen our conclusion that BR

signalling plays an important role in LRC development.

The claim that BZR1 has "LRC-specific" expression (line 124) is misleading; it is expressed throughout the root and more strongly in the columella (but not the LRC). Expression scale differences in the root atlas also suggest redundancy with BZR1 homologues is likely. I.e. although the color gradient is similar for all the expression schemes, the scale is not: whereas the darkest color equals an expression of approximately "1" for BZR1, the same color indicates "1.6" for BES1 or "0.3" for BEH4, etc. This does not take away anything from the fact that the authors observe a matching phenotype in the *bzr1* loss of function mutant (the reference #17 for this mutant is incomplete btw), but it indicates that redundancy with BZR1 homologs is still possible.

We thank the reviewer for raising this important point. We agree that BZR1 is not "LRC-specific," and we have rephrased this section of the manuscript accordingly (lines 214–216). Regarding the differences in expression scale in the root atlas, we have now standardized the visualization by setting the same maximum threshold for all *BZR1/BES1/BEH* genes. Specifically, we set the maximum threshold to 1.7 for all genes, corresponding to the highest value observed for *BES1* (1.69), to allow direct comparison among them. In Figure EV3A, we have added a colour scale indicator on the right and included a more detailed description in the figure legend. To address the potential redundancy among *BZR1/BES1/BEH* proteins in controlling cell number in the L2 layer, we have added confocal images in Figure EV3B–D of the reporter lines *pBEH3::gBEH3-eYFP* and *pBEH4::gBEH4-eYFP*, which we generated in our laboratory, alongside the *pBZR1::BZR1-YFP* line. These data show that only the BZR1 protein is detectable in the L2 layer.

For CYCD3 fluorescence quantification, was the same line used and crossed into other backgrounds, with confirmed equal transgene copy numbers? Otherwise, results from fluorescence quantification could be misleading.

We thank the reviewer for this comment. We have introduced the *pCYCD3;3::GFP* reporter by crossing the *pCYCD3;3::GFP* line (10.1016/j.cub.2014.07.019) into relevant backgrounds. To clarify this point, we revised the description in line 280-282, line 287-289 and line 354-355 of the manuscript.

The use of Welch's t-test raises questions about variance or sample size equality.

We have changed our statistical analysis for all the data concerning cell number quantification and cell division events, please refer to the corresponding comment raised by Referee 3. Further we added all details on statistical tests performed and sample size in the figure legends.

Driver line insertion sites, if known, should be stated.

We thank the referee for this comment and have added the details on the GAL4/UAS driver lines *J1092* and *J3411* that were analysed in ([10.1104/pp.16.00213](https://doi.org/10.1104/pp.16.00213)) to the methods section.

typos/grammar:

line 25: "self-renewing"?

line 70: "programmed"?

line 82: "in all cells" / "even in neighbouring" we rephrased as in line 176-177

lines 105-109: rephrase for clarity we have added a to the introduction section

line 154: "(SFig. 2C-F)"?

We thank the referee for spotting typos/grammar and have addressed all in the revised manuscript.

Referee #2:

This study examined the role of brassinosteroid (BR) signaling in controlling the proliferation of lateral root cap (LRC) cells in *Arabidopsis thaliana* roots. Using a series of fluorescent marker lines and genetic tools, including Timer-NLS, the authors demonstrated that a specific domain of the L2 layer of LRCs is the center of BR-

mediated cell proliferation control, where *CYCD3;3* plays a significant role in cell cycle progression. The study extends our understanding of the role of BRs in root meristem maintenance with higher spatial resolution. To improve the readability and scientific accuracy of the paper, I ask the authors to address the following questions. We thank the referee for the encouraging feedback on our manuscript.

1. Specificity of BR signaling modulation

I wonder whether the promoters used to drive *bzr-1d* and *bin2-1* are sufficient to conclude that L2d is the site of BR signaling for LRC proliferation. Both the FEZ promoter and the *J1092* driver are active outside of the L2 domain. Also, to conclude the specific role of BR signaling in the L2, the cell numbers in the L1 and L3 domains should also be quantified after the modulation of BR signaling.

We thank the reviewer for this important question. Prior to using the FEZ promoter and the *J1092* driver, we extensively searched for L2-specific marker lines; however, both in our hands and according to the literature, these remain the best currently available driver lines.

To strengthen our conclusion that BR signalling acts primarily from the L2 layer, we added a quantitative analysis of the L1, L2, and L3 layers in the *bri1-116 bri1 bri3 (bri³)* mutant (Fig. 2A,B) and in the *J1092>bin2-1* and *J1092>bzr1-D* lines (Figure EV3J,K). These analyses show that the primary defects originate in the L2 layer. Importantly, alterations in L2 length are accompanied by corresponding changes in the L3 layer in the *bri³* mutant and in the *J1092>bin2-1* line. In contrast, the increased L2 size observed in *J1092>bzr1-D* does not lead to a proportional increase in L3 size.

Together with the fact that cell divisions occur predominantly in the L2 layer and the expression pattern of *CYCD3;3*, these data support our conclusion that BR signalling operates mainly from the L2 layer and secondarily impacts L3 to control cell number in the LRC.

2. Relationship with global root meristem (RM) regulation:

BRs are known to regulate root meristem activity and SCN maintenance. The data in Fig. 1A confirm that global modulation of BR synthesis and signaling strongly impacts RM size. Considering this, it is important to analyze whether local manipulation of BR signaling in the LRC affects RM size. If RM size remains unaffected, the significance of the postulated local LRC proliferation control becomes clearer.

We thank the reviewer for raising this interesting point. To assess whether altered BR activity in the *J1092* domain affects the root meristem, we quantified root meristem size in *J1092*, *J1092>bin2-1*, and *J1092>bzr1-D* lines (Figure EV3L). We found that *J1092* and *J1092>bzr1-D* display comparable root meristem sizes, whereas the *J1092>bin2-1* line exhibits a significantly shorter root meristem. This observation is consistent with the reduced L3 layer length in the *J1092>bin2-1* background and aligns with our previous findings showing that the LRC contributes to the control of root meristem size (Di Mambro et al., 2019).

3. Scientific accuracy and data quality

Please carefully check the accuracy of the introductory paragraph. The authors should clarify that the description is for *Arabidopsis thaliana* roots. The explanation of the root meristem and SCN is somewhat mixed up. The number of cells in the L1-L3 layer refers only to those appearing in a longitudinal section of cell files. Each layer is composed of tens of times more cells around the circumference. The *CYCD3;3* reporter fluorescent signals are too weak (Fig. 4A), and I question the reliability of the image quantification.

We thank the reviewer for raising this important point. We have added an introductory paragraph specifying that our descriptions refer to *Arabidopsis thaliana* roots, and we have included a more detailed description of the SCN and LRC tissues (line 28-43).

Regarding the *pCYCD3;3::GFP* reporter fluorescence signal, we thank the reviewer for drawing our attention to this issue. A careful examination of the confocal image quality revealed that, during file import into Adobe Illustrator, the colour mode had been inadvertently set to CMYK. This resulted in artifacts when importing RGB images, which were particularly evident in confocal images using green–magenta channels. We have corrected this issue, and all confocal images in all figures have now been properly imported in RGB format.

Miscellaneous

The text annotations in most figure panels are generally too small and have low color contrast. The graphs are also too small. Please consider labeling the L1-L3 layers in all figure panels where possible. There is mislabeling of fluorescent proteins in Figure 3A and of the X-axis in Figure 3B.

We thank the referee for the comment and apologize for the small graphs. We have now set all graphs according to the EMBO Reports min. size of 7 pt solving this problem. We also added a labelling of the L1-L3 layers in all Figure panels where possible. We thank also for spotting the mislabelling in the Fig. 3A and 3B.

The definition of distal and proximal LRC cells is somewhat counterintuitive, considering the organ-level morphology. It would be better to mention why such a definition was adopted.

We thank the referee for noticing this important issue. We agree that our definition of distal and proximal L2 cells was not in alignment with the organ-level morphology, therefore we have now adopted in all figures and in the text the classical organ-level definition, referring distal as towards the root tip and proximal towards the shoot. We specified this definition in the introduction (line 29-31) and applied this change also to the results and discussion section accordingly.

Figure 4A: If the statistical test ends with pairwise comparisons, why did the authors perform an ANOVA (Figure 4B)?

We thank the reviewer for the comment. We performed a one-way ANOVA because more than two groups were analysed. We performed multiple comparisons using a Tukey's post hoc test to compare each cell with each other. Only biologically relevant pairwise comparisons are shown in Figure 4A.

Throughout the manuscript, the term "cell division activities" would be better replaced with "cell numbers" or "cell proliferation capacities" and so on, as the authors did not measure the division activities per se.

We agree with the referee and have changed throughout the manuscript the term "cell division activities".

Line 154, SFig. 3C-F should be SFig. 2C-F.

Thank you for spotting this error.

Please thoroughly check the English writing.

We thank the reviewer for this comment. We have thoroughly revised the manuscript to improve the English writing throughout the text.

Referee #3:

In this exciting manuscript, the authors link BR-signalling to cell division activity of the root cap cells. The manuscript is well written, easy to read and all experiments follow in a logical order and are well designed to answer the questions at hand. I find this manuscript to contain a very interesting finding which will be of interest to a broader plant developmental biology public. There are some remarks and suggestions below to further improve the manuscript, some of which are not required, but would be nice to include in case the data would be available. The remarks regarding the statistics and inter-ecotype crosses would be the most important to address from my point of view.

We thank the referee for the positive feedback.

Line 41: if the TIMER-NLS line changes from fade blue to red, why do we only see blue nuclei in Figure1?

We used the fade-blue Timer-NLS signal as an indicator of recent cell divisions in plants, as described in Gaillochet et al., 2016 (doi:10.7554/eLife.30135). We also attempted to image the mature red fluorescent Timer (FT); however, we observed the mature red FT in nearly all cells, which is in accordance with the reported half-life of the mature red FT, which is according to Subach et al., 2009 (doi:10.1038/nchembio.138) in HeLa cells persisting for 63–80 hours. In contrast, the Timer-NLS has the advantage that the "fade-blue" FT form marks cells that have undergone division recently, within the last 20–36 hours. This time window, based on Gaillochet et al., 2016, is ideal for quantifying the most recent cell division activities in plants.

Line 44: please include the penetrance of the phenotypes mentioned in text also on the figures for clarity.

We thank the reviewer for this suggestion and have added in the figure legend of Fig. 1 D the penetrance of the cell division events observed in the L3 layer.

Line 48: although the usage of the timer-NLS line is a nice proxy, it provides a static image of the divisions occurring in the root meristem. I am wondering why the authors did not try to catch the divisions in a time-series experiment (as is now done routinely in many labs) where one could generate a much better quantification of these events. I am not claiming this is crucial for the message the authors are bringing, but it would provide stronger quantitative evidence in case this is feasible for the authors to perform. As an alternative to the dynamic imaging, it would be good to complement the timer-NLS line by e.g. a well-known pCYCB1;1-DB-GUS/GFP line as the destruction box would also allow to highlight only the dividing cells (it is not clear to me if the line in Fig1G has a DB or not. If so, this is solved, but the authors should make this clear). My point being that it would be good to have more than a single marker to illustrate this process as it is at the basis of the manuscript.

We thank the reviewer for noticing this important point. We employed the *pCYCB1;2::CYCB1;2-GFP* translational reporter (Boruc et al 2010), which includes the DB and is subject to rapid degradation during anaphase. Thus, in the experiment of Fig. 1 G-H and in Figure EV. 2 the cells CYCB1;2-GFP mark cells in the late G2-to-M transition, marking only actively mitotic cells. We have added a correct reference of the line and a description of its use to mark only actively dividing cells in the Line 141-145.

Line 58: "However, BR mutants also display defects in the LRC tissue". This statement needs some references to support it.

We thank the referee for noticing this, the reported evidence are extremely indirect. Hence, we rephrased as follow: "BR regulate meristem size and cell shape by acting cell non autonomously from the LRC (Ackerman-Lavert et al., 2021; Nolan et al., 2023). However, whether BR signalling acts also cell autonomously within the LRC is unexplored."

Line 72: "We noticed an increased size of the L2 after treatments with BL, supporting that this hormone promotes LRC size": can this be indicated on the figure as it is perhaps not as clear to the untrained eye. Similar remark for Fig 2D and E.

We thank the reviewer for this suggestion. We now marked in all figures the different layers of the LRC with L1, L2 and L3.

Line 83: "the expression domain of J2632 was slightly expanding ". Slightly is not a very scientific term. The signal intensities in Fig 2C-E should all be quantified before making this statement based on one image. Please include the quantification in the figure to support these statements.

We thank the reviewer for this important point. We apologize for using slightly and removed this word. Instead, we added in line 178-179 a more detailed description of the domain, since our objective in the experiment of Fig. 2C-D was to analyse domains, not intensities within the LRC.

Line 89: Semantic discussion: the statements are made about the action of BR, but are the authors sure it is BR itself or BR-signaling which is doing this? Given this is unknown at this point in the manuscript, perhaps the statements should be about the exogenous treatment of BR to the root and not the action of BR as such.

We thank the reviewer for this comment and agree that this point was not clear in the previous version of the manuscript. We have now included the analysis of the *br1-116 br1 br3* triple mutant in Figure 2A–B, which provides strong evidence that BR signalling is indeed involved in determining cell number in the LRC. Since the statement regarding cell division using the two markers was at this point in the manuscript under exogenous BR treatment, we have revised the text accordingly (lines 194–196) as suggested.

As a more general remark relevant throughout the manuscript, the use of a t-test or ANOVA for discrete data-points (such as cell number quantifications where you only have discrete numbers: 1, 2, 3 etc; and not 1.3, 2.4, 4.5 etc) is not the best option as the data is by definition not normally distributed, but discretely. I would suggest the authors contact a statistician to apply a more correct statistical test in these cases throughout the manuscript. It would be important to ensure all results are still significant when applying the correct statistics.

We thank the reviewer for raising this important issue. We have consulted two experts for statistics that suggested us to apply for analysis of discrete numbers such as for cell number and cell division events by applying nonparametric tests. Therefore, we applied the Mann-Whitney U test for two-group comparisons and the Kruskal-Wallis test followed by Dunn's post hoc test for multiple comparisons. All tests were two tailed and $p < 0.05$ was considered significant. We re-analysed all the data with discrete numbers and results remained still significant.

Fig 3A: panels show both YFP and CFP fusion, but the legend only mentions CFP. I am assuming the YFP in the lower panel in A should be CFP.

Thank you for spotting this, we have corrected the panel A.

Line 121: 'genesand' should be 'genes and'

Thank you for spotting this typo.

Line 122: "We took advantage of the single cell root atlas ", perhaps more correct to write "of a single cell atlas" as there are many published.

We have added 'a single cell atlas'.

Line 124: looking at these predicted expression patterns, I would say none of these genes has a root cap specific expression. BZR1 is also predicted to be in the cortex. Some, if not all, of these genes are well studied. Are no markers or expression patterns published for these?

We thank the reviewer for this comment and agree that these genes are not specific to the LRC but instead show broader expression patterns. While markers for BEH genes have indeed been published, they were unfortunately not shared despite request. In the meantime, we have generated translational reporter lines for BEH3 and BEH4 in our laboratory and have included confocal images of the root meristem in Figure EV3C–D.

Line 141: "Hence, we generated *UAS::bzip1*D plants and crossed them with the J1092 enhancer trap line". Given the lines like J1092 were originally generated in the C24 ecotype, does this mean you made an inter-ecotype cross with Col-0? If so, this is known to have very striking effects on root growth and comparison to the original line in C24 is not valid. One would then need to make a control cross (J1092 x Col-0) and compare to this one. In case all was introgressed into Col-0 background, this is not required; but I did not see any info in the materials and methods regarding this. Same remark for the other crosses made involving enhancer trap lines.

We thank the reviewer for this important comment. We have transformed C24 plants with the *UAS::bzip1-D* and *UAS::bin2-1* constructs and subsequently crossed the selected lines with the J1092 driver line. Thus, all experiments using J1092 driver lines were performed in the C24 background, and we have added this information to the Materials and Methods section (line 378-380).

In contrast, the *pFEZ::GVP* line is in the Col-0 background, and we crossed it with our *UAS::bzip1-D* and *UAS::bin2-1* lines in the C24 background, resulting in a Col-0 × C24 hybrid background. In these lines, we compared mock-treated and dexamethasone-treated plants within the same line to control for background effects.

Line 159: although not required, it would be nice if the authors could validate the large scale ChIP-ChIP experimental results for the BZR-CYCD3 interaction themselves. As I understand this is not a trivial experiment, it would be nice to add this data in case it is available.

We thank the reviewer for the suggestion. The BZR1–CYCD3;3 interaction has also been identified in an independent ChIP-seq experiment (doi:10.7554/eLife.03031; see Figure EV4A). Therefore, we believe that the combination of two independent ChIP experiments (one ChIP–chip and one ChIP–seq) showing BZR1 binding to the *CYCD3;3* promoter, together with our genetic evidence supporting the functional interaction of BZR1 and *CYCD3;3* in LRC development, strongly suggests that BZR1 directly regulates the *CYCD3;3* gene.

Dear Dr. Unterholzner

Thank you for the submission of your revised manuscript to EMBO reports. It has been seen again by former referee #1 who considers all concerns adequately addressed and supports publication.

Before I can accept the manuscript, I need you to address some minor points below:

- Please provide a 'Disclosure and competing interests statement'. For more information see <https://www.embopress.org/page/journal/14693178/authorguide#conflictsofinterest>

- The information on funding provided in the manuscript and in the online submission system must be congruent. Please correct the following:
Ministero dell'Università e Ricerca PRIN2022 grant (Grant number 20229LHY5L) - grant number not matching in the system and the paper;

PIANO NAZIONALE DI RIPRESA E RESILIENZA (PNRR) - MISSIONE 4 COMPONENTE 2, INVESTIMENTO 1.4 - D.D. 1032 17/06/2022, CN00000022 - not listed in the submission system.

- The manuscript sections should be in the following order: Title page - Abstract & Keywords - Introduction - Results - Discussion - Methods - Data Availability - Acknowledgments - Disclosure Statement & Competing Interests - References - Figure Legends - (Main Tables with legends if applicable) - Expanded View Figure Legends.

- Highlights need to be removed from the manuscript

- Please address the following comments in the figure legends:

1) Provide the exact p values in the legends of figures 2B, F, H; 3B, D, F, H, J; 4B, D, F, H; EV1 C, D; EV2 B, EV3 G, I, K, L; EV4 D, F, H

2) Define the box plots in terms of minima, maxima, centre, bounds of box and whiskers, and percentile in the legends of figures 1C, 2B, F, H; 3B, D, F, H, J; 4B, D, F, H; EV1 B-D; EV2 B, EV3 G, I, K, L; EV4 B, F, H.

- You re-analysed ChIP-chip experiments from (Sun et al., 2010) and (Oh et al., 2014). You could use the format of Data citations, where you first cite the primary paper and then in addition the dataset (Sun et al, 2010; data ref: Sun et al, 2010). Data citations in the article text are distinct from normal bibliographical citations and should directly link to the database records from which the data can be accessed. In the main text, data citations are formatted as follows: "Data ref: Smith et al, 2001" or "Data ref: NCBI Sequence Read Archive PRJNA342805, 2017". In the Reference list, data citations must be labeled with "[DATASET]". A data reference must provide the database name, accession number/identifiers and a resolvable link to the landing page from which the data can be accessed at the end of the reference.

- I did a spot check on the supplied source data, as we always do. Are you sure that Snap-64_1.tiff is the correct source data image for Figure 1B? Moreover, in several folders you supply more than one image, are these from replicate experiments? It might be helpful to include a little README.txt file in the folders to explain this.

- Moreover, the source data need to be reorganized to one file/folder per figure and ZIPing for each main figure. For EV and/or appendix figures, ZIP together all source data.

- Finally, EMBO Reports papers are accompanied online by

A) a short (1-2 sentences) summary of the findings and their significance,

B) 2-3 bullet points highlighting key results and

[Please provide this text as a separate file]

C) a schematic summary figure that provides a sketch of the major findings (not a data image).

Please provide the summary figure as a separate file in PNG or JPG format at a size of 550x300-600 pixels (width x height).

Please note that the size is rather small and that text needs to be readable at the final size. Please send us this information along with the revised manuscript.

With kind regards,

Martina

Martina Rembold, PhD

Senior Editor
EMBO reports

=====

Referee #1:

I would like to thank the authors for their careful revisions. From my perspective the manuscript is ready to be published in its current form.

The authors addressed the remaining editorial issues.

Dr. Simon Unterholzner
Free Universtiy of Bolzano
Faculty of Science and Technology
Piazza Università 5
Bolzano, BZ 39100
Italy

Dear Dr. Unterholzner,

I am very pleased to accept your manuscript for publication in the next available issue of EMBO reports. Thank you for your contribution to our journal.

You may qualify for financial assistance for your publication charges - either via a Springer Nature fully open access agreement or an EMBO initiative. Check your eligibility: <https://link.springer.com/journal/44319/how-to-publish-with-us>

Yours sincerely,

>>> Please note that it is EMBO Reports policy for the transcript of the editorial process (containing referee reports and your response letter) to be published as an online supplement to each paper. If you do NOT want this, you will need to inform the Editorial Office via email immediately. More information is available here: <https://link.springer.com/partners/embo-press/editorial-policies#Peer%20review>